physiology/evolution/health and disease and epidemiology

hatching failure, female fertility, reproduction, egg production, sperm storage, fertilization

**Author for correspondence:**
Katherine Assersohn
e-mail: kassersohn1@sheffield.ac.uk

# Physiological factors influencing female fertility in birds

Katherine Assersohn[1], Patricia Brekke[2] and Nicola Hemmings[1]

[1]Department of Animal and Plant Sciences, University of Sheffield, Sheffield S10 2TN, UK
[2]Institute of Zoology, Zoological Society of London, Regents Park, London NW1 4RY, UK

 KA, 0000-0002-1085-0266; PB, 0000-0001-6298-3194;
NH, 0000-0003-2418-3625

Fertility is fundamental to reproductive success, but not all copulation attempts result in a fertilized embryo. Fertilization failure is especially costly for females, but we still lack a clear understanding of the causes of variation in female fertility across taxa. Birds make a useful model system for fertility research, partly because their large eggs are easily studied outside of the female's body, but also because of the wealth of data available on the reproductive productivity of commercial birds. Here, we review the factors contributing to female infertility in birds, providing evidence that female fertility traits are understudied relative to male fertility traits, and that avian fertility research has been dominated by studies focused on Galliformes and captive (relative to wild) populations. We then discuss the key stages of the female reproductive cycle where fertility may be compromised, and make recommendations for future research. We particularly emphasize that studies must differentiate between infertility and embryo mortality as causes of hatching failure, and that non-breeding individuals should be monitored more routinely where possible. This review lays the groundwork for developing a clearer understanding of the causes of female infertility, with important consequences for multiple fields including reproductive science, conservation and commercial breeding.

## 1. Introduction

Fertility is fundamental to reproductive success, and so we should expect fertility traits to be under strong selection to maximize reproductive output and minimize the wastage of gamete investment [1,2]. Despite this, fertility varies remarkably across individuals, species and populations [3,4], and some degree of infertility is ubiquitous across taxa. Gametic wastage is likely to be more costly for females than males [4], since they typically invest

considerably more in gamete production [5]. These costs are also likely to be greater in taxa that produce large yolky ova, such as birds. Despite this, females are thought to have received comparatively less attention than males within avian fertility research [6], especially in poultry research, where attempts to increase fertility have historically been focused on male performance with far less attention given to females [7,8].

Successful fertilization occurs when a male pronucleus and a female pronucleus fuse to form a zygote (i.e. syngamy) [9]. We therefore define infertility as the failure of syngamy, and any male or female process contributing to failed syngamy is a cause of infertility. Confusingly, the term infertility has been used interchangeably in the literature to describe both fertilization failure and embryo mortality across taxa, though these two processes often have a very different mechanistic basis [10]. Differentiating between infertility and embryo mortality is often difficult, particularly if the embryo dies during the very early stages of development [11], but the failure to distinguish between them presents an important barrier to addressing the underlying causes of reproductive failure.

Birds are well suited to the study of reproductive failure, primarily because—unlike mammals—they produce large, well-protected eggs which make them easy to examine externally before they degrade [11]. It is also possible to precisely determine whether a bird's egg failed because it was unfertilized or because the embryo died very early, using microscopic methods, but many studies still fail to make the distinction [10,12,13]. Avian reproduction science also benefits from a wealth of knowledge from commercial poultry research. Despite intensive, long-term selection for consistent and efficient egg production in certain lines of commercially important species such as the domestic fowl (*Gallus gallus domesticus)* and turkey (*Meleagris gallopavo)*, hatching failure is still a pervasive issue in many commercial breeds, and the reasons for this are not fully understood [11,14]. In the wild, hatching failure is also ubiquitous; on average 10% of eggs never hatch [2], and in some threatened and bottlenecked species more than 60–70% of eggs fail [15,16]. While there has been some attention paid to embryo mortality in birds, we still lack a clear understanding of the incidence of true infertility and the factors that contribute towards it [17]. The incidence of infertility relative to embryo mortality in wild populations has most likely been overestimated by many studies [11], while in captive birds, infertility may be more likely [10]. Understanding the mechanisms that cause infertility could therefore be particularly important for captive breeding programmes.

Here, we provide a thorough review of female-specific physiological factors that lead to infertility in birds. We have consolidated the most valuable insights from across a broad range of literature, including behavioural ecology, evolutionary biology and reproductive physiology, integrating these with key findings from the vast but often under-cited poultry science literature. We reveal that there has been a deficit of fertility research in wild and non-commercial species, and that female fertility traits are consistently understudied relative to those of males. We then identify and explore key phases in the female reproductive cycle where fertility may be compromised, and broadly categorize the physiological mechanisms of infertility into five key processes: (i) failure to produce fertilizable eggs, (ii) failure during ovulation, (iii) failure to obtain sufficient sperm, (iv) failure to store and transport sperm, and (v) failure of fertilization. We draw particular attention to the relationships between senescence, environmental factors and female reproductive function. Our aim is to develop a clearer understanding of the proximate causes of variation in female fertility and highlight key directions for future research.

## 2. How much do we know about female fertility traits in birds?

We conducted a systematic search of the avian fertility literature (see electronic supplementary material for methods), identifying 718 relevant papers on avian fertility traits, of which 42% considered both male and female fertility, 37% focused on male fertility only and 20% focused on female fertility only. As expected, the number of avian fertility papers published each year is increasing, but since 1985, the number of published papers that focus on male fertility have increased at a faster rate than the number of papers focused on female fertility (figure 1*a*) ($x^2 = 15$, d.f. = 2, $p < 0.001$). By April 2020, published papers focusing on male avian fertility outnumbered those on female avian fertility by a factor of 1.84, indicating that there is a deficit of papers focusing on females (compared with males) within avian fertility research, a gap that appears to be widening over time. However, studies that considered fertility traits in both males and females were almost as numerous as those that considered males only, perhaps indicating that many researchers are taking a more holistic approach. This may reflect the inherent difficulties involved in disentangling the effects of male and female factors on fertilization success, since they are likely to be non-independent processes exhibiting complex interactions [18]. Across all years (from 1921 to 2020), 79% of articles exclusively investigated captive populations, with only 16% investigating wild populations and 5% investigating both captive and

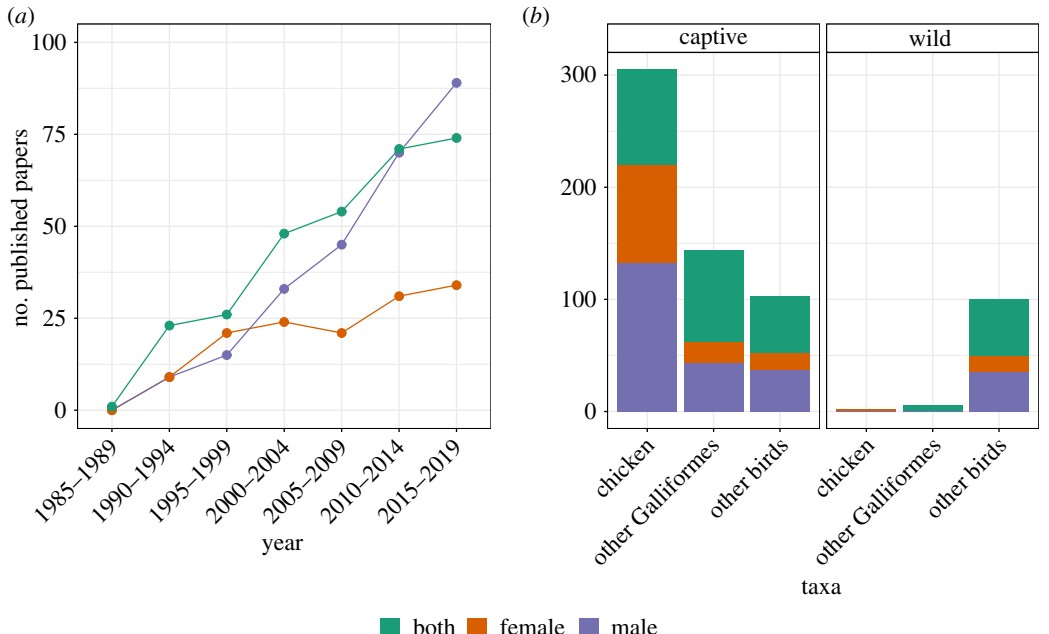

**Figure 1.** (a) The number of papers on avian fertility published between 1985 and 2019 on males only (purple), females only (orange) and both males and females (green). (b) The number of papers on avian fertility published between 1921 and 2020 (all years) either focusing exclusively on the domestic chicken (*Gallus gallus domesticus*), other Galliformes, or other (non-Galliform) bird orders, for both captive (left) and wild (right) populations.

wild populations. Furthermore, of the captive species studied, 80% focused on the order Galliformes, with 54% focused exclusively on a single species: the domestic chicken (*Gallus gallus domesticus*). This indicates that the avian fertility literature has also been heavily dominated by studies on gallinaceous birds of commercial importance, with far fewer studies investigating non-commercial and wild species. The greater focus towards male fertility appears to be a consistent pattern across both captive and wild populations (figure 1b). Within wild populations, only 15% of papers focus on females relative to 32% that focus on males. Within captive populations, the greater focus on male fertility is not solely driven by studies in poultry, since it is consistent even when domestic fowl are excluded, in which case only 14% of studies focus on females relative to 33% that focus on males.

The deficit of female-only avian fertility papers may reflect that, relative to ova, it is more practical to study sperm traits, partly because it is easier to collect sperm in a way that is non-invasive and repeatable. In birds, sperm biotechnology has also been developed considerably in poultry over the last century [19], which has advanced the ways in which sperm can be examined, preserved and manipulated. There may also be a degree of positive feedback, with ease of collection and study of sperm yielding greater advances in methodology, which in turn yields further research. Beyond the practical advantages to studying sperm relative to ova, the over-representation of studies on male fertility may also be a consequence of the historical view that sperm are the 'active' participants in fertilization: seeking, binding to and penetrating the somewhat 'passive' egg [20]. Cultural biases have also been suggested to drive a male-orientated research focus across other taxa including mammals [21]. While this view has been challenged in recent years (especially with regard to post-copulatory processes such as cryptic female choice [22]), the gap between the number of male and female fertility papers (both for wild and captive populations) suggests that female fertility traits in birds are still understudied, and the role of the female in determining reproductive success is therefore underappreciated. The following sections explore the physiological mechanisms that may contribute to variation in female fertility in birds, and suggest new hypotheses and future directions that may help fill the gaps in our current understanding of avian female fertility.

# 3. What causes infertility in female birds?

## 3.1 Failure during egg formation

In birds, infertility is typically measured as the number of unfertilized eggs, but this makes the assumption that a female is already able to produce an ovum that can be fertilized. Female fertility is

the product of not only fertilization rate, but also the number of eggs produced that are capable of being fertilized. The process of egg formation—from follicular development through to release of the ovum during ovulation—is a metabolically demanding process [23,24] and problems occurring during these early stages of reproduction are a costly and important cause of infertility. The incidence of infertility resulting from egg production problems is unknown for most wild bird populations, making it difficult to determine how important it is as a driver of individual variation in fitness. Regardless, determining the incidence of egg production dysfunction is a logical step towards establishing the causes of reproductive failure. Future studies should (where possible) attempt to collect data on failed breeding attempts (i.e. where copulation was successful, but no eggs were produced), particularly when the fertility status of the male breeding partner is known. This is likely to be somewhat easier for captive populations relative to long-term wild study populations, where data are not routinely collected on non-breeding individuals. Much of the avian fertility literature has a heavy focus on seasonally breeding species, where photoperiod provides a reliable cue eliciting an annual reproductive response. Generally, less is known about tropical/a-seasonal species and opportunistic breeders, where an individual's ability to respond rapidly to more unpredictable environmental cues is likely to significantly influence their fertility.

Reproductively active females of most bird species have only one functional oviduct in which there is usually just one ovary (figure 2). The right oviduct regresses during development, and this is thought to occur via hormonally controlled apoptosis (caused by the release of anti-Müllerian hormone), while the left oviduct is protected from regression by elevated concentrations of oestrogen (which inhibits the anti-Müllerian hormone receptor) [25], although the molecular mechanisms underpinning this process are not yet fully defined. The mature avian ovary contains multiple maturing follicles each at a different stage of development [26] (labelled $F_1$–$F_5$ in figure 2). Mature follicles consist of a large, protein rich yolky oocyte and a small germinal disc (which contains the genetic material), surrounded by a granulosa cell layer, multicellular theca layer and an epithelial layer (figure 3) [27,28]. At the vegetal pole of the follicle, the epithelial layer becomes thin, forming a region known as the stigma that acts as the point of follicle rupture during ovulation [28]. During the later stages of follicular growth, a glycoprotein structure known as the perivitelline layer forms between the granulosa cells and the oocyte (figure 3). The perivitelline layer functions to bind with sperm during fertilization and initiate the acrosome reaction [29].

In birds, the mechanisms underpinning follicular development, maintenance and selection have yet to be well defined, although the implications of these processes are likely to be significant for fertility [30]. Follicular selection (i.e. the selection of one white follicle to rapidly uptake yolk, undergo further differentiation and eventually ovulate as a mature yellow follicle) is thought to be mediated by cyclic adenosine monophosphate (cAMP) signalling, which acts through G protein-coupled receptors to upregulate the expression of multiple genetic factors important for follicular development [25,30–32]. The unselected white follicles are maintained in an undifferentiated/arrested yet viable state within the ovary until the next follicular selection [28]. This is thought to be regulated in part by the β-arrestin protein, which desensitizes G protein-coupled receptors (and thus inhibits cAMP signalling), and depresses granulosa cell differentiation [30]. Understanding the mechanisms governing follicular recruitment and maintenance is considered a primary challenge in avian reproductive research [28,33].

### 3.1.1 Hormonal factors

The proper functioning of the avian endocrine system is vital for egg production. In seasonally breeding species, photoperiodic cues are received by deep brain photoreceptors that stimulate activity of the hypothalamic-pituitary-gonadal (HPG) axis. The HPG axis is a tightly regulated system that, among other things, regulates physiological processes associated with reproduction [28]. Specifically, following an increase in photoperiod, the mediobasal hypothalamus is stimulated to produce local thyroid hormone which regulates the release of gonadotrophin releasing hormone. This in turn stimulates the pituitary to produce gonadotropins that initiate seasonal gonadal growth and activity [28,34]. Following breeding, the HPG axis is promptly 'switched off', resulting in a significant regression of the gonads [28]. In male Japanese quail (*Coturnix japonica*), lesions of the mediobasal hypothalamus can inhibit the photoperiodic response and gonadal growth [35], but whether such lesions affect seasonal gonad development in females is unclear.

The degree to which endocrine disorders naturally affect wild birds is largely unknown, but disorders such as cystic hyperplasia, cystic ovaries and hypothyroidism are a clinical issue in captive birds [36–40]. There is also extensive experimental evidence showing that hormonal disruption can significantly

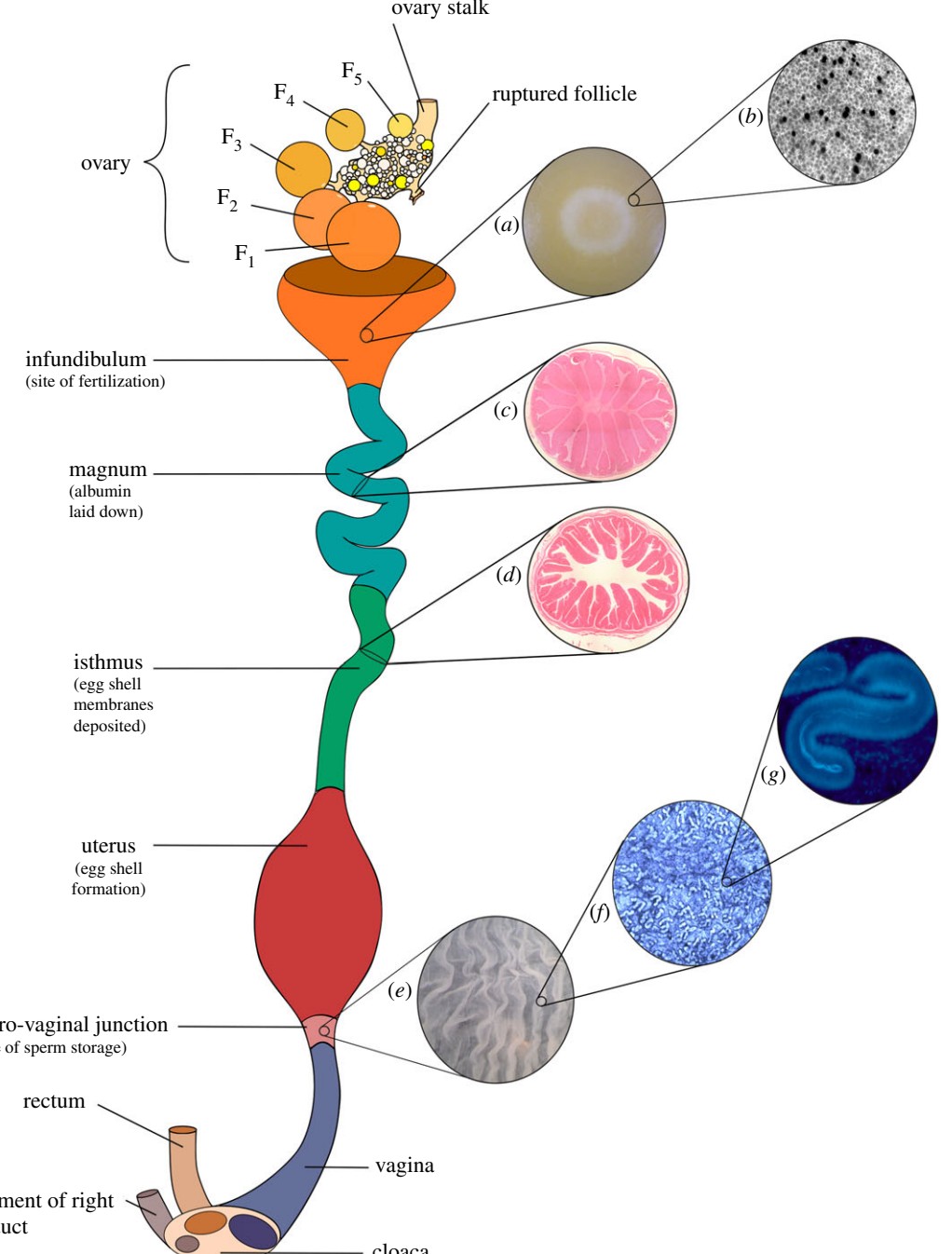

**Figure 2.** Schematic of the avian oviduct and ovary (not to scale). Note that the avian ovary consists of multiple follicles at different stages of development. The largest yellow follicles are labelled $F_{1-5}$ where $F_1$ is the largest follicle and will be the next to rupture. (*a*) The germinal disc of a fertilized ovum (from a zebra finch (*Taeniopygia guttata*)). Note the clear outer ring and paler center of the germinal disc which indicates embryonic development. (*b*) Sperm penetration holes visible on the inner perivitelline layer (IPVL) of an ovum after fertilization (from a bullfinch (*Pyrrhula pyrrhula*)). (*c*) A cross section of the magnum (from a helmeted guineafowl (*Numida meleagris*)). Sperm is transported through the magnum prior to fertilization, but this region functions mainly to produce the albumin which is laid down during egg development. (*d*) A cross section of the isthmus (from Reeves pheasant (*Syrmaticus reevesii*)). Sperm is also transported through the isthmus prior to fertilization, but this region functions mainly to produce and deposit shell membranes during egg development. (*e*) The internal tissue lining and folds of the vagina and utero-vaginal junction region (from a bobwhite quail (*Colinus virginianus*)). The vagina is considered the primary site of sperm selection in the oviduct, and the utero-vaginal junction functions as the primary site of sperm storage, containing numerous sperm storage tubules. (*f*) A single fold of the utero-vaginal junction (from a zebra finch (*Taeniopygia guttata*)) stained with Hoechst 33342 dye under a fluorescence microscope. The many small tubular structures are sperm storage tubules. (*g*) A single sperm storage tubule (and visible trapped sperm) from a single fold of the utero-vaginal junction (from a Japanese quail (*Coturnix japonica*)), stained with Hoechst 33342 dye and viewed under a fluorescence microscope. Image credits: (*a–d*) Nicola Hemmings; (*e–g*) Paul Richards.

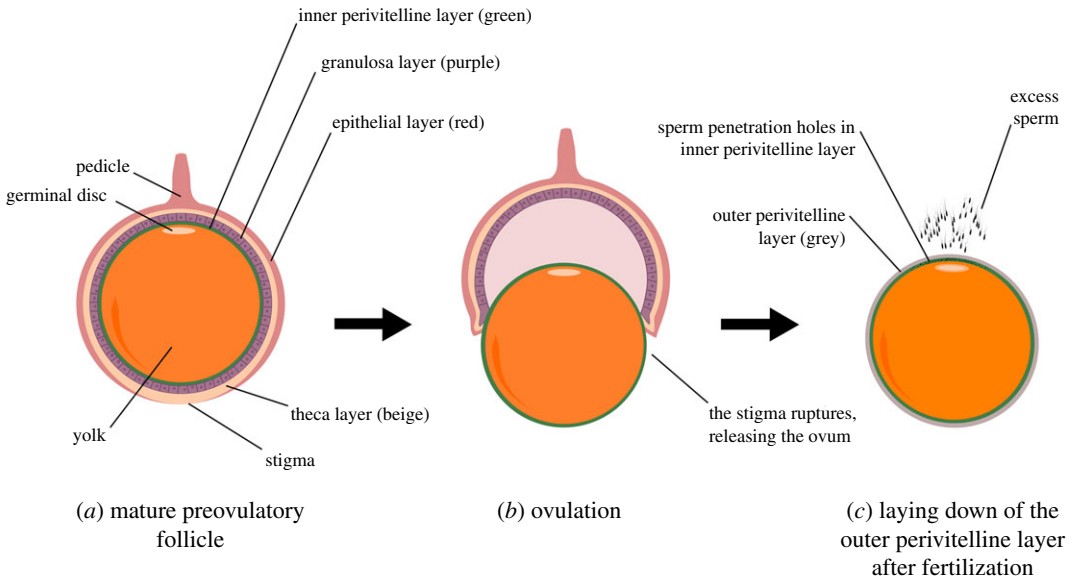

**Figure 3.** Schematic of a mature avian follicle. (*a*) An avian follicle prior to ovulation. (*b*) The ovum and follicle during ovulation, whereby a mature follicle ruptures at the stigma region, releasing the ovum. Sperm present in the infundibulum will begin to move towards the ovum in preparation for fertilization, where they will penetrate the IPVL (green). (*c*) The ovum after fertilization, the outer perivitelline layer (OPVL) (grey) has been laid down (which blocks further sperm entry). The IPVL (green) has an abundance of sperm penetration holes around the germinal disc region where sperm have penetrated during fertilization (figure 2*b*).

influence egg production in captive species. For example, in domestic fowl, administering luteinizing hormone 8.5 h after ovulation causes follicular degeneration (atresia) of the next follicle within the follicular hierarchy [41], and in pigeons (*Columba livia*), administering synthetic gonadotropin-releasing hormone can reduce luteinizing hormone concentrations, depressing egg production [42]. Treatment with the inhibin A protein can increase the proliferation of granulosa cells and increase the secretion of granulosa steroid production, while decreased expression of the inhibin $\alpha$ subunit (which has been observed in cystic follicles in pigs [43]) is associated with follicle atresia in chickens [32]. Counterintuitively, injections of follicle stimulating hormone can decrease egg production in zebra finches (*Taeniopygia guttata*), possibly because of a negative feedback effect with endogenous secretion of follicle stimulating hormone [44], although timing of treatment is also likely to be important since the function of follicle stimulating hormone is known to vary with follicle size under normal conditions [28]. Inappropriate levels of the anti-Müllerian hormone have been shown to disrupt normal reproductive development in some species [45,46], and experimentally inhibiting oestrogen synthesis in chicken embryos increases the expression of the anti-Müllerian hormone receptor, resulting in masculinization of the reproductive tract [47]. The anti-Müllerian hormone is thought to play a vital role in follicular development within the ovary, and elevated levels have been associated with periods of restricted fertility in hens [48]. Elucidating the full significance of the anti-Müllerian hormone for avian female fertility is an active area of research [33,48].

### 3.1.2 Disease and immune factors

The proper functioning of the immune system is of great importance in the defence against bacterial, fungal and viral pathogens within the ovary. Toll-like receptors (TLRs) produced in the follicular tissue of domestic fowl are known to be involved in the recognition of pathogens, and play a key role in inducing an innate immune response in the ovary [28]. In particular, TLRs respond to pathogenic stimuli by producing avian β-defensins (antimicrobial peptides), and proinflammatory cytokines and chemokines [28,49]. Other cellular members involved in the innate immune response include macrophages, natural killer cells and antimicrobial peptides [28]. TLR signalling may also cause disruption to steroidogenesis, and result in apoptosis of undifferentiated granulosa cells, thereby providing a mechanism to prevent the selection of infected follicles into the preovulatory hierarchy [50]. The adaptive immune response then involves the migration of certain immunocompetent cells

into the follicles, including major histocompatibility complex (MHC) antigen-presenting cells, T cells, B cells and macrophages [28]. The distribution of immunocompetent cells in the oviduct increases during sexual maturation, but then decreases significantly thereafter with age [51]. In humans, ovarian autoimmunity has been associated with premature ovarian failure and infertility [52]. Less is known about the incidence and mechanisms of ovarian autoimmunity in birds (particularly wild populations), but antibodies that target ovarian tissue have been identified and been associated with a decline in egg production with age in laying hens [51,53], and autoimmune thyroiditis is a clinical issue associated with obesity and fertility declines in chickens [40]. The frequency of immunocompetent cells present in the follicles of the hen ovary also decreases with age, suggesting a reduction in infection resistance in older hens that could have an associated impact on egg production [54]. The follicular reserve also depletes as birds age [55], inevitably resulting in changes to the HPG axis. This may occur via reduced secretion of gonadal steroids and peptides and/or reduced sensitivity of the hypothalamus to ovarian steroids, either because of diminished steroid stimulation or a general pattern of neural senescence [55]. Oestrogen has been associated with the upregulation of immunocompetent cells into maturing follicles, and so may be involved in the age-related decline of the immune response in the ovary [28,51,54].

In humans, ovarian disorders such as hormonal dysfunction, ovarian abnormalities (such as polycystic ovarian disease), premature menopause and genetic defects [56], explain 30% of female infertility cases. Although less well studied in birds, ovarian disorders are known to reduce or stop egg production in poultry [57], and cystic ovarian disease is common in other bird species (e.g. cockatiels (*Nymphicus hollandicus*), budgerigars (*Melopsittacus undulatus*) and pheasants (*Phasianus colchicus*) [37]) [38]. Microbes present in the intestines and cloaca may be transported to—and colonize—the ovaries [28]. Inflammation of the oviduct (such as salpingitis or metritis) caused by bacterial or viral infection can result in oviductal impaction, egg abnormalities and infertility [38]. Little is known about the incidence of such ovarian disorders, or the degree to which they explain variation in female fertility in wild populations.

### 3.1.3. Environmental factors

### 3.1.3.1 Diet

A wealth of experimental evidence in birds (mostly poultry) shows that diet strongly influences egg production and fertility. Striking the balance between optimum nutrient uptake (to maximize production) and nutrient toxicity and/or obesity is of primary concern to the poultry industry [58]. Modern broiler breeder hens are particularly sensitive to overfeeding during the weeks prior to laying, and even minor overfeeding can result in oviducal inflammation, prolapse and a reduction in egg production [59]. When broiler breeder hens are fed ad libitum, this can result in obesity and the onset of erratic oviposition and defective egg syndrome (EODES), which is thought to be caused by excessive follicle development and the occurrence of multiple follicular hierarchies which disrupts ovulation [59]. Yolk formation is energetically demanding and requires substantial changes in the body's metabolism of lipids [60]. Lipogenesis is responsive to both hormonal control as well as dietary changes, and overfed hens exhibit symptoms of lypotoxicity including ovarian abnormalities and follicular atresia [60,61]. Feed restriction is a commonly used method for controlling obesity and the onset of EODES [59,60]. Over the last few decades, intense selection for greater body mass has resulted in an increase in food consumption during ad libitum feeding. Consequently, there has been an increase in the use and intensity of feed restriction regimes in broilers [58,60,61]. It is becoming increasingly difficult for poultry breeders to achieve a diet sufficient for growth and reproductive maintenance without overfeeding. Restricted feeding protocols also come with additional welfare issues, namely an increase in stress and social aggressiveness related to hunger [62].

Nutrients thought to be important for egg production and fertility in birds include: manganese, selenium, iodine, fluoride, sodium, zinc, copper, vitamin A, vitamin E, vitamin $B_{12}$, protein and linoleic acid [58,63–68]. An excess or deficiency in any of these can be disruptive. Nutritional deficiency and toxicity are common in captive birds but thought to be more rare in wild populations [63], although the dietary requirements and nutrient availability for wild populations are less well studied and may be impacted by environmental change and/or supplemental feeding. In particular, the nutritional needs of endangered populations with reduced natural habitat may be restricted, especially if they have been translocated to habitats with different food sources to those in their native range (but see Jamieson [63]).

### 3.1.3.2 Stress

It has long been known that stress also plays a vital role in the productivity of laying hens, and stressors may include fear (either of humans [69] or of novel social or physical environments [70]), insufficient space [71] and heat stress [72]. Heat stress reduces egg production by decreasing feed intake and causing nutritional deficiencies, but also by causing widespread disruption to the hormones important for ovulation [72]. In birds, temperatures above 30°C can trigger heat stress [73], and Deng *et al*. [74] found that when laying hens were exposed to 34°C heat for two weeks, egg production decreased by 28.8%. Increasing environmental temperatures predicted under climate change is expected to have important repercussions for commercial egg production [73,75]. Evidence for thermally induced female fertility loss in non-commercial species is lacking [76], although in male zebra finches exposed to 30°C and 40°C heat there was an increase in the production of abnormal sperm [77]. Regarding wild populations, it is likely that small, isolated or endemic species are particularly vulnerable to heat stress-induced reductions in fertility, because lack of gene flow and genetic variation impose limitations on their ability to adapt to novel environmental stress. Species with limited ranges may also be at risk if they are unable to shift to cooler climates [76].

Early-life/developmental conditions and stress may also heavily influence individual patterns of reproductive ageing. For example, rates of reproductive senescence are higher in guillemots that invest more heavily in early-life reproduction [78]; the effects of early-life stress (in the form of predation pressure) increases the rate of reproductive senescence in barn swallows (*Hirundo rustica*) [79]; and female collared flycatchers (*Ficedula ablbicollis*) from a low-competition natal environment experience higher reproductive rates early in life, but with a cost of earlier reproductive senescence [80]. In seasonally reproducing wild birds, older females with previous breeding experience often (initially) lay earlier in the season, and lay larger clutches than inexperienced females [81–83]. This has been attributed to females having prior experience responding to photostimulation, resulting in more robust photo-induced reproductive development, including having higher circulating levels of reproductive hormones and greater seasonal increases in follicle size than in photo-naive birds [81,82]. Initial increases in reproductive output with age are often followed by gradual declines in fertility with age across species [83]. In the endangered and bottlenecked whooping crane (*Grus Americana*)—known for suffering from fertility problems—female age is a key predictor of egg fertility, where younger females have a greater probability of producing fertile eggs [84]. Fertility senescence can be very variable across species, however, and while some short-lived species (such as poultry) experience rapid declines in female fertility with age, many long-lived birds apparently experience little or even no reproductive decline throughout their lives [83,85]. We do not fully understand the mechanisms of fertility senescence in birds [79]; more long-term longitudinal studies on age-related changes in fertility traits are necessary, especially in wild (non-poultry) species and where environmental/ developmental effects are incorporated [78,80,83,86].

### 3.1.3.3 Pollution

Exposure of wild birds to environmental pollutants has been shown to have a significant impact on fertility [87]. Over 90 000 anthropogenic chemicals are estimated to have been released into the environment, and several hundred of these pollutants are confirmed to act as endocrine-disrupting compounds (EDCs; although the majority remain untested for their effects on wildlife) [88,89]. If passed on to developing embryos, EDCs can disrupt reproductive development and cause sterility [90]. Negative effects on reproductive success in birds have been observed even when they are exposed to very low and environmentally relevant concentrations of certain EDCs. For example, small amounts of crude oil are sufficient to depress egg yolk formation in seabirds [91,92]; exposure to flame-retardant additives at concentrations typically seen in the environment can depress reproductive success (including fertility) in American kestrels (*Falco sparverius*) [93], and in areas polluted with environmental oestrogens, severe reproductive tract abnormalities have been found in exposed females [92]. Toxic heavy metals are also known to act as EDCs and disrupt reproduction in exposed birds: a single dose of cadmium was enough to significantly reduce egg production in Japanese quail [94]; lead is known to accumulate in the ovaries of pheasants [95], Japanese quail [96] and chickens [97] following exposure (which can reduce egg production and cause histopathological damage and developmental delays in the ovaries); and exposure to mercury (even at very low concentrations) can significantly reduce reproductive success in zebra finches [98]. EDCs have also been shown to disrupt mating behaviour in several seasonally breeding birds [99] and are linked with population declines [100]. Understanding how EDCs influence reproductive physiology in wild birds is crucial,

particularly for endangered birds where even small reductions in fertility can jeopardize species survival [101]. Relatively few long-term studies of wild birds have monitored the effects of EDCs on avian fertility, and in particular there is a lack of knowledge on the population-level effects of EDCs on fertility in wild birds [102,103]. Detecting the varied and often sublethal effects of EDCs is difficult: wild birds are probably exposed to many different types of EDC at one time [104,105], each with potentially complex and different effects [103]. The risks of EDC exposure to fertility is also likely to differ between species and across individual lifetimes [106]. Identifying the mechanisms by which EDCs affect fertility in wild birds therefore requires a combination of laboratory studies, long-term monitoring, and continued development of analytical methods [103].

## 3.2 Failure during ovulation

Ovulation occurs when the largest mature yellow follicle (labelled $F_1$ in figure 2) ruptures at the stigma region (figure 3) [28], releasing the ovum which is then captured by the infundibulum—the site of fertilization. Unlike mammals, the granulosa layer provides the main source of gonadal steroids [32], and ovulation is initiated by the production of testosterone in the granulosa cells, which stimulates the release of granulosa cell progesterone. Progesterone then creates a positive feedback response in the hypothalamus which stimulates an increase in the secretion of gonadotropin-releasing hormone, and consequently causes a surge of pituitary luteinizing hormone [31,107,108]. Clock genes expressed within granulosa cells after follicle selection are thought to provide a degree of circadian control over the timing of ovulation [31,109]. Proper regression of the post-ovulatory follicle is also thought to be required for managing the timing of ovulation and egg-laying [25].

### 3.2.1 Hormonal factors

As with egg formation, hormones play a critical role in regulating the process of ovulation. [25]. Exposure to environmental EDCs is likely to disrupt normal ovulation in exposed birds [110], and experimental hormonal manipulation can significantly influence ovulation (and therefore fertility) in captive species. For example, an increase in progesterone at the wrong time may induce a spike in luteinizing hormone, which triggers premature ovulation [38,111]. Inhibition of luteinizing hormone (e.g. via serotonin injections) [111] can lead to anovulation and disruption of thyroid hormone function [65], and ovulation can be prevented in domestic fowl when treated with the testosterone antagonist flutamide, which blocks the production of preovulatory hormones [112].

### 3.2.2 Disease and immune factors

In captive pet birds, excessive ovulations are one of the most common forms of reproductive abnormalities [113], the causes of which are not fully understood, but may be improved with husbandry and dietary measures [114]. Coelomitis is another common clinical problem in domestic birds, causing inflammation of the ovaries (oophoritis) and ectopic ovulation [37]. Viral infections such as avian influenza, infectious bronchitus and avian hepatitis can cause the formation of chronic lesions within the oviduct, that may prevent the successful capture of ova following ovulation [115,116]. This can result in extensive damage to the oviduct [117], often leading to further bacterial infection due to the presence of yolk in the coelomic cavity (egg yolk peritonitis) [118]. The spontaneous development of ovarian cancers is extremely common in the laying hen [28], the incidence of which increases with age, occurring in 24% of hens aged over 2 years [119], and 30–35% of hens by 3.5 years [25]. The increase in the incidence of ovarian cancers with age is thought to be, at least in part, a consequence of the accumulation of ovarian surface and DNA damage caused by ovulatory events over time [120]. Laying hens may therefore be at particular risk from ovarian cancers because of the selection for frequent ovulations in commercial breeds [28]. Progesterone can be effective at reducing the incidence of ovarian cancers, possibly because it limits the number of ovulations experienced [121]. Increased levels of progesterone have also been implicated in an increase in the number of apoptotic events in the ovary, which may act to remove damaged cells [122].

In broiler breeder hens which have been selected for rapid growth at the expense of fertility, double-yolk eggs are fairly common, and occur more frequently during the onset of egg production [25,123]. Double-yolk eggs are associated with a greater incidence of embryo mortality (at all stages of development) and are also more likely to be infertile [123], possibly because ova are ovulated early and in an immature state. Ovulation order of double-yolk eggs also affects the likelihood of

fertilization: in duck (*Anas platyrhynchos domesticus*) eggs, the first yolk captured by the infundibulum has a higher probability of being fertilized [124]. This may explain why double-yolk eggs commonly contain only one fertilized ovum [123]. Age, nutrition (e.g. feed restriction) and changes in photostimulation are all thought to play a role in the production of double-yolk eggs, the occurrence of which can also be increased via selection [124], indicating a genetic component.

## 3.3 Failure to obtain sperm

Even if ova are produced and ovulated normally, sperm must have been obtained and be present in the infundibulum at the precise time of ovulation. In the majority of bird species (greater than 97%), sperm are obtained by the female through cloacal contact during copulation, although there are a few examples of extant bird species that copulate with the use of a true intromittent organ (e.g. in the ratites, many Tinamiformes, Anseriformes and Cracidae), or a pseudo-phallus (such as in some Galliformes) [125–127].

### 3.3.1 Copulation

Sperm may be prevented from reaching the site of fertilization in several ways. Mechanical difficulties during mating resulting from physical injury (e.g. impaired vision or balance [38]) may prevent sperm from entering the reproductive tract. Access to the cloaca (figure 2) may be physically blocked due to obesity or clogged feathers (e.g. due to fecal build-up, or heavy cloacal feathering) [39,128,129], though this may be more likely to occur in captive populations. In captive birds, failed copulation may also occur due to inappropriate husbandry, for example a lack of proper perching or nesting sites, aviary disturbances, a lack of flock stimulation or illness [39]. Immaturity and sexual inexperience may also result in failed mating in young birds [39]. If mating proceeds normally, then females could theoretically ensure sufficient sperm are available for fertilization by copulating more frequently. Using an experimental approach that restricted inseminations by males, Török *et al*. [130] showed that multiple copulations were necessary to achieve a normal (unmanipulated) level of egg fertilization success in wild collared flycatchers, implying copulation frequency is important for fertility assurance [131]. However, multiple copulations could also reduce fertility if they damage the female reproductive tract. Copulation provides an opportunity for bacterial transfer [132,133], which can cause local inflammation in the vaginal wall, impairing sperm transport and reducing fertility [134]. When artificial insemination is performed with poor technique, the risk of vaginal infections is increased [135,136]. In domestic turkey hens, inflammatory effects of repeated artificial insemination appear to be transient, with quick recovery [137], but the long-term consequences of repeated infections are unknown. The main defence against microbial infection in the oviduct is provided by the vaginal mucosa and associated mucin substances, although cilial action within the oviduct may also play a role in the removal of microbes [28]. Similarly to the ovary, TLRs and avian β-defensins are also expressed within the rest of the oviduct, as well as macrophages, natural killer cells, cathelicidin and other antimicrobial defensins such as gallin [28]. There is a high density of T cells and B cells in the vagina [138], and the expression of proinflammatory cytokines are also increased in response to oviductal infection [139].

### 3.3.2 Timing of insemination

If ovulation proceeds normally, the ovum progresses into the infundibulum and a second glycoprotein layer is formed around it within approximately 15 min, preventing additional sperm from penetrating (figure 3). This short fertilization window requires precise timing of insemination and/or the release of sperm from female storage to ensure sufficient sperm are in the infundibulum at ovulation [140]. In turkeys, sperm storage is more efficient when artificial insemination is performed just prior to (rather than just after) the onset of egg production [141]. Similarly, in chickens, more sperm reach the infundibulum when they are inseminated within 1–6 days of ovulation (relative to 6–8 days before ovulation) [142]. The greater the time delay between insemination and ovulation, the fewer sperm will be available for fertilization, since sperm are lost passively from storage at a constant rate [143]. The time interval between ovulation and egg-laying takes approximately 24–28 h in chickens, turkeys and quail [144]. Inseminations performed immediately before or after egg-laying reduce fertility, possibly because egg-laying contractions impede the ability of sperm to move through the oviduct and/or the passage of sperm is blocked by the egg [26,28]. In chickens, sperm storage is up to 40 times more

efficient when insemination occurs more than 4 h after egg-laying [134]. Similar evidence of low sperm uptake during and just after egg-laying has also been observed for natural copulations [145].

### 3.3.3 Vaginal sperm selection

Ensuring sufficient sperm are available for fertilization has clear benefits for the female, but the mechanisms that might facilitate this are at odds with those facilitating female sperm selection. The vagina is considered the main sperm selection site in the avian oviduct [146,147], and only 1% of inseminated sperm make it through to the sperm storage tubules (labelled ($f$) and ($g$) in figure 2). Domestic fowl, for example, eject the sperm of undesirable males following forced copulations [148], thereby reducing the number of sperm available for fertilization. The vaginal fluid of female barn swallows has also been shown to reduce sperm performance by varying degrees depending on female quality [149]. Huang *et al.* [150] found that vagina mucosal tissue of chickens produces exosomes (membrane vesicles enriched in transmembrane proteins) that significantly reduce sperm viability (possibly because they contain cytotoxic factors) and therefore may play a role in sperm selection. During egg production, vaginal pH and immunological activity also varies [26,138], and immune-competent cells are expressed in the vagina [151]. Van Krey *et al.* [152] found that infertile females express antibody-producing plasma cells in their reproductive tract, and Higaki *et al.* [153] showed that the number of leukocytes present in the vagina increases following copulation. Localized immune responses are predicted to participate in non-random sperm selection [154], and the likelihood of an anti-sperm response may depend on male genotype [155]. For example, Løvlie *et al.* [156] show that post-copulatory sperm selection in female red junglefowl (*Gallus gallus*) is biased towards males dissimilar at the MHC. Sperm are rapidly coated with immunoglobulin cells produced by the vaginal mucosa, and immunoglobulin IgA and IgG are thought to be at least partially responsible for the massive reduction in sperm viability during transport through the vagina [26]. Determining how female anti-sperm responses vary across individuals is a crucial step towards understanding the importance of the immune response for avian female fertility. Immune response strength probably depends on complex phenotypic trade-offs between fertility and infection resistance [157], similar to the trade-off females face between fertility and sperm quality during sperm selection. If sperm selection and/or immune response mechanisms are too effective, insufficient sperm may reach the site of fertilization. Sperm selection and transport in the vagina may therefore be considered a balance between selecting high-quality sperm, avoiding infection, and ensuring sufficient sperm remain available for fertilization.

## 3.4. Failure to maintain and transport sperm

### 3.4.1 Sperm storage tubule function

Within the female reproductive tract, sperm are stored in blind-ended tubular invaginations known as sperm storage tubules (SSTs) [147] (labelled ($f$) and ($g$) in figure 2) found in the utero-vaginal junction (UVJ). While SSTs are considered the primary sperm storage site, it has been suggested that sperm may also be stored in the infundibulum. However, evidence for this is equivocal [154]. Once sperm are in storage, the proper functioning of the SSTs is likely to be crucial to fertility, and a decline in sperm storage ability has been associated with fertility senescence in birds [158]. The number of sperm stored is strongly associated with the number that reach the ovum [141], and chickens selected for high fertility have significantly greater numbers of SSTs [159]. The mechanisms controlling sperm acceptance, storage and release are assumed to be under fine temporal control based on hormonal changes [142]. Fertilization failure is therefore likely if hormonal imbalances result in a mismatch between the timing of ovulation and arrival of sperm in the infundibulum [142,154]. The significant variation in fertile periods across species has been attributed to differences in SST number and therefore sperm storage capacity [154], but intra- and inter-specific variation in the structure and function sperm storage tubules has not yet been quantified.

The mechanisms by which sperm are maintained in a viable state in the SSTs are not fully understood, but it is thought that numerous compounds are produced to maintain a suitable environment for long-term sperm survival [154,160]. Recent work provides some experimental evidence to support this. For example, lactic acid—now known to be produced in SST cells in response to hypoxic conditions—can induce a reduction in sperm flagellar movement and so may contribute to the quiescence of Japanese quail sperm [28,161]. Additionally, Huang *et al.* [162]

identified a number of fatty acids in the UVJ mucosa of domestic fowl, which have also been shown to depress rooster sperm motility, and *in vitro* sperm survival was found to be higher in the presence of oleic and linoleic acid. They also found that SST cells express lipid receptors, which may enable lipid droplets to accumulate and be used by resident sperm to maintain structural integrity. SST cells in turkeys are also known to shed microvillus vesicles that interact with sperm, transferring metabolic substrates that may be capable of temporarily inhibiting fertilizing ability, protecting from oxidative stress and transporting fluid from the SST cells into the SST lumen [163]. While evidence suggests that these compounds play a key part in sperm maintenance, there are probably a number of other important compounds expressed in SSTs that have yet to be discovered [28]. The degree to which individuals and species vary in their ability to maintain sperm in storage is not clear, but this may play an important role in determining female fertility.

One factor that is essential for sperm survival in storage is the suppression of the female immune response, which if triggered can have highly detrimental effects on sperm. In domestic fowl, repeated artificial inseminations were associated with a complete lack of stored sperm and a 57% decrease in fertility [164,165]. This has been attributed to an influx of lymphocytes and antigen-presenting cells into SSTs that probably impair sperm survivability, but also prevent sperm from entering storage [164]. A significant decrease in the expression of oestrogen receptors in the sperm storage tubules was also observed following infection, probably impairing the hormonal control of sperm storage tubule function [165]. Das *et al.* [166] demonstrated enhanced local expression of transforming growth factor *β*(TGFb) within sperm storage tubules, which suppresses the anti-sperm immune response by depressing the activity of lymphocytes.

### 3.4.2 Sperm release and transport

As the rate of sperm release from storage increases, the duration of fertility will decline unless more sperm are inseminated [26]. Older hens tend to release sperm faster, possibly due to a decline in hormone production needed to regulate ovulation and sperm release [167]. This may partially explain why older hens have shorter fertile periods [26]. The mechanisms of sperm release are not fully understood, but SSTs have been shown to possess a constricted 'gate-like' entrance, that may act as a physical (and/or selective) barrier preventing sperm from leaving [168]. Constriction is likely to be hormonally triggered, since progesterone has been shown to induce contractions of the SSTs, and images taken using electron microscopy show sperm leaving the SSTs after intravenous injection with progesterone [169]. Furthermore, the specific membrane progestin receptor mPR$\alpha$ has been shown to be expressed within SSTs of Japanese quail [169]. Additionally, in a comprehensive study, Hiyama *et al.* [170] demonstrated that heat shock protein 70 (HSP70)—a widespread and highly conserved molecular chaperone—is expressed in the utero-vaginal junction and its expression increases prior to ovulation. They also found that HSP70 binds to sperm and stimulates flagellar movement *in vitro*, and injection of an HSP70 antibody significantly reduces fertilization success *in vivo*. Hiyama *et al.* speculate that HSP70 expression in the UVJ may be stimulated by progesterone; the progesterone surge experienced prior to ovulation may therefore function in part to allow sperm to be released from storage at the right time while also ensuring sperm regain function. Imbalances in circulating progesterone levels can be triggered by conditions such as nutrient toxicity (e.g. excess fluoride [68]) and heat stress [171], and future work should explore if such imbalances reduce fertility by disrupting sperm release from storage.

Once released from storage, sperm are thought to travel passively through the uterus, isthmus (labelled (*d*) in figure 2) and magnum (labelled (*c*) in figure 2) [154], as evidenced by the fact that dead sperm inseminated beyond the utero-vaginal junction reach the infundibulum in as great numbers as live sperm [172]. Past the utero-vaginal junction the reproductive tract is apparently free of immunoglobulins, and anti-peristaltic activity is thought to aid in the passive and rapid transport of sperm to the infundibulum, [173] where they remain until fertilization [154].

## 3.5. Failure of sperm–egg fusion

Following successful ovulation, the avian ovum is captured by the infundibulum where it encounters sperm (figures 2 and 3). Successful fertilization involves the initiation of multiple events in sequence: sperm–egg binding, acrosomal exocytosis, sperm penetration through the perivitelline layer and fusion of the male and female pronuclei in the germinal disc. The mechanisms of sperm–egg interactions in birds are not well understood, but the roles of several important molecules have been discovered.

### 3.5.1. Sperm–egg interactions

The inner perivitelline layer (IPVL) (figure 3), which is homologous to the zona pellucida (ZP) in mammals, is composed of a mesh of fibre that forms a three-dimensional extracellular matrix. Unlike in mammals, the IPVL of birds does not inhibit polyspermy [29], and in fact a degree of physiological polyspermy is required for normal development in birds [140]. There are at least six known avian ZP glycoproteins [6,17] (there has been significant confusion in the literature regarding the nomenclature of ZP proteins [174], here we provide the common aliases in parentheses), most notably ZP1 (ZPB1), ZP3 (ZPC) and ZPD which are major components of the IPVL and play a key role in the binding of sperm and initiation of the acrosome reaction [17,29]. Other minor constituents of the IPVL include ZP2 (ZPA), ZP4 (ZPB or ZPB2) and ZPAX (ZPX1) [28,175], where ZP2 accumulates primarily in the germinal disc region in chickens [176]. Interestingly, ZP4 (but not ZP2) mRNA was found to be expressed in the germinal disc region in the turkey [6], suggesting differences in sperm binding mechanisms may occur across species. Acrosin, located in the sperm plasma membrane, was discovered to be a complementary molecule that supports the binding of sperm to the ZP proteins in quail PVL [17].

Diagrammatic representations of fertilization often depict the ovum oriented such that the germinal disc faces towards the reproductive tract (and oncoming sperm). However, we suggest that the animal pole—where the germinal disc is located—faces towards the ovary during ovulation and fertilization (figure 3), because the ovum is in that orientation while inside the follicle [28], and to our knowledge there is no known mechanism by which it would turn to face the opposite direction after ovulation. If true, the consequence is that sperm would have to travel around the ovum to reach the tiny germinal disc target. Sperm are known to bind to the germinal disc region in higher concentrations than elsewhere on the ovum [176], suggesting there must be an underlying mechanism by which sperm locate and/or preferentially bind to this region. Such a mechanism has yet to be discovered, but the egg plasma membrane is reported to differ in morphology around the germinal disc region compared with other regions of the ovum [177]. Specifically, numerous microvilli and cytoplasmic processes of the plasma membrane have been observed to protrude through the IPVL exclusively at the germinal disc region [177–179]. In other (non-germinal disc) areas of the ovum, the discontinuous nature of the ovum plasma membrane has been hypothesized to inhibit sperm from hydrolysis over these regions [180]. It has also been suggested that sperm may locate the germinal disc region via egg chemo-attractants [181], and/or by using site-specific egg coat receptors [176]. PVL glycoproteins ZP2 and ZP4 are promising egg coat receptor candidates, since ZP2 and ZP4 are concentrated primarily in the germinal disc region of chicken and turkey PVL respectively, but their sperm binding properties have yet to be investigated [6,17,176]. Recently, a number of new PVL proteins have been identified which appear to vary across species [182], but their function in fertilization has not yet been determined.

In mammals, a variety of protein coding genes associated with gamete cell surfaces have been discovered [183–185]. This includes Juno and Izumo—the only known interacting pair of sperm–egg adhesion proteins. Izumo is a sperm protein [186], and Juno is the more recently discovered egg Izumo receptor [187]. In mice, Juno and Izumo knockouts result in sterility, and Juno is vital in preventing polyspermy; its rapid loss from the egg surface membrane following fertilization causes the blocking of the zona pellucida to further sperm entry [187]. Reproductive proteins are known to evolve rapidly compared with many other gene classes, and both Juno and Izumo have been found to be under positive selection in mammals [188]. No such interacting proteins have been discovered in birds: comparisons of the genomic regions containing Juno (and surrounding loci) in mice and humans with that of the chicken shows that they are generally syntenic (the gene order is conserved); however, Juno loci are absent in the chicken [28]. A key step for avian fertility research will be to identify avian Juno and Izumo equivalents. Also essential for sperm–egg fusion is the ubiquitously expressed membrane protein CD9: female (but not male) CD9 knockout mice are infertile [185]—CD9 was the first identified gene with female-specific fertility effects [189]. There is one known homologue of CD9 in the chicken (ID: AB032767) though to our knowledge no studies have explored the involvement of this (or any other gene) on sperm–egg fusion in birds [28].

In addition to proteins, the consistency and structure of the IPVL differs markedly between species. For example, Damaziak et al. [182] observed that cockatiels have a more densely and irregularly arranged IPVL than that of three other species studied (pigeons, grey partridges (Perdix perdix) and pheasants). They also found that the pigeon PVL is markedly different in structure; its numerous sublayers are more homogeneous, less porous and unusually loose in arrangement compared with the other species. Pigeon PVL is also composed of flat sheets rather than the cylindrical fibres which are observed in the PVL of all the other species. It is unknown how this variation in structure affects the

integrity of the PVL, its interaction with sperm, or whether this variation corresponds to post-copulatory sexual selection intensity and/or sperm traits. Damaziak *et al.* [182] suggest that interspecific variation in PVL structure may be related to differences in the function of the PVL during embryo development, which may vary depending on whether the species is precocial or superaltricial. The germinal disc region is also known to show subtle intra- and inter-specific variation in terms of morphology, and also in terms of the location, size and number of sperm penetration holes [190]. It is currently unknown how variation in PVL structure affects fertility, but fertilization rates are positively correlated with the number of sperm that penetrate the PVL [191]. There is known to be variation in how readily sperm can bind to the PVL [146,192], and it seems logical that intra- and inter-specific differences in PVL structure may affect how easy it is for sperm to bind to and penetrate the PVL. In turkey lines where females are selected for increased body weight at the expense of fertility, fewer sperm penetration holes are visible on the IPVL compared with hens selected for high fertility [193]. While fewer IPVL sperm holes could indicate that fewer sperm are reaching the site of fertilization, it has also been associated with reduced mRNA expression of ZP1 and ZP3 sperm binding proteins on the IPVL [8]. This not only suggests that individuals may vary in terms of the number of sperm that are able to penetrate the IPVL, but also that selection for higher expression of sperm binding proteins may improve fertility in some populations [8,193]. In taxa where polyspermy is lethal to the egg (such as in mammals), egg 'fertilizability' is known to vary according to the risk of polyspermy [194]. Consequently, females and males will be locked in an apparent cycle of coevolutionary conflict where females are selected for greater 'egg defensiveness' (resistance to sperm) and males selected to counter this with greater fertilizing ability and competitiveness [21]. Currently, egg defensiveness has been most largely explored in sea urchins and in mice [22,194], with virtually nothing known in birds. Since polyspermy is a normal and important part of fertilization in birds, this suggests that mechanisms of polyspermy avoidance are unlikely to be important other than to prevent excessive sperm penetration that might damage the integrity of the ovum. Theoretically, females might be expected to evolve mechanisms of resistance to sperm for other reasons, for example, to alleviate the costs of hybridization, avoid incompatible sperm, or as a mechanism of selection for high-quality sperm [22,175]. Indeed, evidence suggests that the strength of positive selection on gamete-recognition genes is similar between birds and mammals, suggesting that in the absence of polyspermy avoidance, there must be some other adaptive mechanism to explain the rapid evolution of avian gamete-recognition genes [175]. Recently, Hurley *et al.* [195] found significant variation in PVL sperm numbers between breeding pairs of estrildid finches, as well as variation in PVL sperm numbers across the laying order. It was unclear, however, if this variation was male or female mediated. Exploring the degree of variation in egg quality, egg defensiveness and sperm selection at the gametic level is challenging but important for elucidating the full role of the avian ovum for fertility.

Chromosomal abnormalities, such as whole genome triploidy, can significantly or completely impair fertility in some affected individuals, and some triploid embryos are non-viable and die after a few days of incubation [196]. Triploidy is usually (but not always) maternally derived, and is thought to arise from diploid gametes produced as a result of chromosomal non-disjunction (where homologous chromosomes fail to separate during meiosis) [197,198]. Reports of triploid birds in the wild are rare (but see [199]), possibly because of the reduced survival of triploid embryos, although the true rate of incidence in wild populations is unknown. In addition to chromosomal abnormalities, the presence of multiple germinal discs on a single yolk has been reported [200], although the cause and incidence of this is unknown, as well as the implications it might have for fertility and embryo development.

Whether other aspects of ovum quality, such as physiological abnormalities, or the integrity of DNA in the female pronucleus, influences the likelihood of successful sperm–egg fusion in birds remains unclear. It seems likely, however, given that in mammals, the accumulation of DNA damage in oocytes is known to impact fertility [201]. Oocytes may be especially vulnerable to the accumulation of DNA damage (relative to sperm) given that they remain in an arrested state for an extended period of time [201]. While oxidative DNA damage is expected to be minimized by various innate protective mechanisms, the efficacy of these mechanisms is known to decline with age [202]. If a similar process occurs in birds, this could be a factor that contributes to fertility senescence in captive species. Increased DNA damage due to oxidative stress has also been associated with exposure to endocrine disrupting compounds in mammals [203], as well as certain reproductive diseases in humans [204]. In mammals, heritable mutations of ZP2 and ZP3 are also known to cause infertility [205], and antibodies raised against ZP proteins can depress ovarian function [6]. Investigating the incidence of similar ovum abnormalities and immunological activity, and the degree to which they might affect avian fertility, may be a fruitful avenue for future research.

### 3.5.2. Syngamy

Once bound with the PVL, the sperm acrosomal contents (including proteases and endopeptidases) are released during the acrosomal reaction and locally degrade the PVL, forming a hole via which sperm can penetrate the ovum (figure 3) [17,28]. Sperm penetration holes are visible on the PVL *in vitro* (labelled (*b*) in figure 2) and can be used as a reliable proxy for the number of sperm that reach and penetrate the ovum [140,195]. Following the acrosomal reaction, the inner acrosomal membrane of sperm becomes exposed, binds to the ovum and the male pronucleus is released. While multiple sperm can penetrate the PVL in birds, only one male pronucleus typically fuses with the female pronucleus in the germinal disc during syngamy. This provides additional potential for the female pronucleus itself to be selective [206], although the exact mechanisms of avian syngamy remain unknown [28]. In the comb jelly (*Beroe ovata*), where fertilization is also polyspermic, the female pronucleus migrates within the egg cell prior to syngamy, moving between different immobile male pronuclei before finally fusing with just one [206,207]. Whether sexual selection occurs at the point of syngamy in birds remains to be seen, but could present an additional opportunity for females to influence the outcome of fertilization by discriminating between males, for example, based on male genotype, or on the integrity of the male's DNA [155,206].

Supernumerary male pronuclei are probably degraded by DNAses in the germinal disc and PVL of mature oocytes [208], though the precise molecular mechanisms involved have not yet been fully elucidated [28]. During or immediately after fertilization, a granular continuous layer is laid down around the ovum, followed by the outer perivitelline layer (OPVL), which blocks further sperm entry (figure 3) [28]. The OPVL is multi-layered and composed of proteins secreted by the infundibular mucosa [209]. Macrophages present within the infundibulum are thought to function in the phagocytosis of superfluous sperm (i.e. those that did not participate in fertilization) [210]. Elucidating the full molecular mechanisms involved in avian syngamy is challenging [28], but will be an important step in understanding how fertility can be compromised at the gametic level.

## 4. Conclusion and future directions

It is clear that females can exert far more control over fertilization than has historically been assumed, but if and how females influence whether their ova are successfully fertilized is often ignored in favour of male processes (such as sperm quality and quantity) [6]. Here we have quantitatively demonstrated that avian fertility research has been dominated by studies on males, with a deficit in research effort on female fertility. We also show that the vast majority of avian fertility research has concentrated on captive populations, with a significant taxonomic focus on gallinaceous birds and the domestic chicken in particular. We have also highlighted key advances and gaps in knowledge on the role of female physiological processes in determining fertilization success. In particular, we have identified five key stages in the reproductive cycle during which fertility can be compromised: (i) failure to produce fertilizable eggs, (ii) failure during ovulation, (iii) failure to obtain sufficient sperm, (iv) failure to store and transport sperm, and (v) failure of fertilization. We highlight that the field of avian fertility would benefit from more studies investigating variation in fertility in non-poultry species (i.e. that have not undergone intense artificial selection for high productivity) and wild populations. Within wild birds, more attention to a-seasonal/tropical species and opportunistic breeders would also be valuable. Although we acknowledge that detailed study of variation in female fertility may be difficult in wild populations, because information about non-breeders is not always easy to collect, we nonetheless urge that such efforts are made, particularly in non-poultry species and managed and/or experimental populations where male processes can be controlled for. If studies are unable to monitor failed breeders (i.e. those that did not produce any eggs following a successful copulation), it would be useful to acknowledge that infertility may be underestimated. Obtaining a more accurate estimate of fertility rates across wild populations, and improving our understanding of the mechanisms that influence fertility, will aid in the management of threatened populations that suffer high levels of hatching failure, and improve predictions for how fertility will be influenced by a changing climate.

When investigating reproductive failure, the fact that infertility and embryo mortality are fundamentally distinct processes needs to be explicitly acknowledged. Specifically, that infertility is used to describe failed fertilization, and any process contributing to failed fertilization is a mechanism of infertility (rather than embryo mortality). Similarly, if an egg fails to hatch but was fertilized, then the cause of hatching failure must be referred to as embryo mortality, even if development arrested

after only a few cell divisions. If fertility status cannot be unequivocally determined using the appropriate techniques [13], then the mechanisms of hatching failure cannot be conclusively known. Very early embryo mortality is likely to be mistaken for infertility when using traditional methods (e.g. candling or macroscopic examination), which may result in an overestimation of infertility [11]. Moving forward, a clearer estimation of the incidence of infertility in a given population will require a combination of both careful monitoring (to identify failed breeders) as well as an accurately determined fertilization status for unhatched eggs.

The female reproductive tract typically offers a hostile environment for sperm, providing considerable potential for female processes to influence sperm survival and transport to the ovum. While the processes of sperm selection, storage, release and transport within the reproductive tract have received increasing research attention over the past few decades, we still lack fundamental understanding of the underlying mechanisms, and the degree of intra- and inter-specific variation in these processes, with the vast majority of work having focused on a very limited number of domestic species. Many of the female-mediated processes required for high fertility also deteriorate to some degree with age, making fertility problems more likely in older birds. This may have particularly important consequences for captive and managed threatened populations, where individuals may reproduce to an older age than their wild counterparts, due to reduced predation and competition pressure, and high accessibility of food and other resources. The field of avian reproductive science will also benefit from better understanding the impact of other factors on female fertility, such as stress, hormonal and physiological disorders (particularly in wild birds where less is known), environmental pollutants, intra- and inter-individual variation in egg production, egg quality, sperm selection and the female immune response within the oviduct (including the ovaries).

The causes and maintenance of variation in fertility is a key question in evolutionary biology, and one in which the role of the female is often sidelined. Our hope is that this review challenges the field of avian reproductive science and evolutionary biology to consider female processes to a greater degree when investigating the causes of depressed fertility in birds. Using birds as a model system for the study of female fertility across taxa presents several advantages and will provide insights not only in the field of reproductive biology, but also for fields such as conservation and commercial animal breeding as well.

Data accessibility. The dataset supporting this article has been uploaded as part of the electronic supplementary material [211].

Authors' contributions. K.A. carried out the data collection, data analysis, created the figures and drafted the manuscript; P.B. critically revised the manuscript; N.H. participated in data collection, assisted in data analysis and critically revised the manuscript. All authors gave final approval for publication and agree to be held accountable for the work performed therein.

Competing interests. We declare we have no competing interests.

Funding. K.A. was supported by the Natural Environment Research Council ACCE Doctoral Training Partnership (grant no. NE/S00713X/1), P.B. was funded by Research England and N.H. was supported by the Royal Society (Dorothy Hodgkin Fellowship DH160200).

Acknowledgements. We thank Karl L. Evans and Ben Hatchwell for helpful and insightful comments on a draft of this manuscript.

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
