## [Peer Review File · Royal Society Open Science]

Review History

RSOS-202274.R0 (Original submission)

Review form: Reviewer 1

Is the manuscript scientifically sound in its present form?

Yes

Are the interpretations and conclusions justified by the results?

Yes

Is the language acceptable?

Yes

Do you have any ethical concerns with this paper?

No

Have you any concerns about statistical analyses in this paper?

No

Recommendation?

Accept with minor revision (please list in comments)

Comments to the Author(s)

I would not consider this to be a comprehensive review. But based on the objectives listed at the beginning of the paper, I am not sure it is meant to be. There are several other papers on the topic in poultry in particular that are not included in this review.

In the sperm:egg interaction section of the review, I know of several papers that are not cited. There is at least one theory of thought that I did not think was included and that is the interaction of the yolk with the IPVL in the non-germinal disc areas of the egg. I provided the authors some of those references but it has been a while since I have reviewed the literature on that topic and I would not consider this to be a comprehensive list.

The rest of my comments and suggestions are in the attached file (see Appendix A).

Review form: Reviewer 2**Is the manuscript scientifically sound in its present form?**

Yes

Are the interpretations and conclusions justified by the results?

Yes

Is the language acceptable?

Yes

Do you have any ethical concerns with this paper?

No

Have you any concerns about statistical analyses in this paper?

No

Recommendation?

Major revision is needed (please make suggestions in comments)

Comments to the Author(s)

I have not followed the avian ovarian physiology literature in several years, but I am knowledgeable in the area of reproductive ecology and behavior. So, I feel I should be able to provide some insight from that perspective. My overall impression of this manuscript is very favorable. It is an important topic, and the authors are correct that fertility problems have been poorly defined historically and that female fertility is a vastly understudied area. The authors seem to cover the topic thoroughly, so it is my opinion that this manuscript would make an important and much-needed contribution to the field. With all that said, however, I am not convinced that Royal Society Open Science is the best venue for its publication, given the journal's own account of favoring papers that are of very broad, general interest. Indeed, I am not sure how broad and general the interest in physiological mechanisms of female bird fertility is. If the paper were on mechanisms of vertebrate or perhaps mammalian fertility, it might be of broader interest. But its current focus on female birds would seem to lend itself better to a bird journal, perhaps, or a journal more specifically oriented toward reproductive physiology, such as *Biology of Reproduction*.

As I indicated, however, I do feel that it is a worthwhile contribution, but some changes, as follows, would make it better. First, it is difficult to understand its overall organization. I wonder if a table of contents would be appropriate. I found it difficult to understand how each section tied in with the one before it and the one after it and then how they all fit together to address the central question: what causes infertility in female birds? One possibility would be to organize the paper around the anatomy of figure 2, walking the reader through each part of egg formation and how fertility can be affected in that part. Additionally, somewhere around line 41, it would be extremely helpful to list as many mechanisms for failed syngamy as possible in order to set up the rest of the paper. The paper could then be better organized around these. Currently, it seems vaguely organized around aspects of Figure 2, but a more deliberate breakdown of the specific ways syngamy can fail, perhaps in order of timing, would greatly facilitate the organization of the paper and therefore understanding it. Currently the paper reads like a long list of what we know about reproduction with little direct tie-in to failed syngamy, especially in any sort of chronological or other logical order.

I think about fertility from an adaptive perspective. Given the presence of a multitude of life-history trade-offs (particularly the trade-offs between number of offspring and quality of offspring within a single brood and also the trade-off between number of current offspring and number of future offspring), perhaps the term "failure" is inappropriate when describing when fertilization does not occur. There may be times when it benefits the female from a fitness perspective to avoid fertilization, even if it means the wasted time and energy that went into production of an infertile ovum or egg. Given that, at least in wild, altricial birds, far more investment occurs in feeding young than in forming the egg for the young, a female suddenly faced with an unexpected challenge (such as unexpected energy demands, predatory risks, or poor quality mate) might benefit by selectively blocking fertilization to avoid costly stages later during nestling rearing when conditions might not merit the investment. For example, the female may be constrained to laying 4 eggs, but challenging food conditions might mean that, this particular bout, 3 eggs might be more adaptive, so she can at least raise 3 good offspring rather than 4 poor ones. In this case, one way to save energy would be to adaptively prevent fertilization of one of the ova, giving rise to 4 eggs but only 3 mouths to feed. Ultimately, I think use of the terms "fail" and "succeed" when it comes to syngamy or fertilization is loaded and assumes fertilization is always adaptive, when there might be situations in which it is not adaptive.

Lines 408-410: Up to this point, the manuscript has mainly focused on what we know in domestic species, each time referring to how little is known in wild species. The reader is left wondering what IS known for wild species. So far, the manuscript mostly reads as a review in domestic species despite what seemed to be a focus early on on the importance of this topic to wild species.

Lines 686-687: It is this possibility specifically that I think of above all else when it comes to possible ways for fertilization to fail, and yet apparently nothing is known. Nonetheless, if the authors could go into greater detail, even if just speculation, I think it would resolve a lot of wondering by their readers.

In several places throughout the manuscript, the authors mention effects of aging on female fertility and mechanisms surrounding it. Although aging is relevant to an extent, it is unlikely to be particularly relevant in wild birds that never get very old. In fact, most wild birds may experience age-related increases in reproductive function during their short lives, and this may be due, in part, to priming by photostimulation (Sockman, K.W., Williams, T.D., Dawson, A., & Ball, G.F. (2004). Prior experience with photostimulation enhances photo-induced reproductive development in female European starlings: a possible basis for the age-related increase in avian reproductive performance. *Biology of Reproduction* 71, 979-986.)

Other comments:

Lines 28-29: Delete.

Line 40: Is this the most broadly used and accepted definition of infertility? If not, reword to say something like "We define infertility as . . ."

Line 87: It may be more technically difficult to study and understand female compared to male fertility.

Lines 133-136: Out of curiosity, how is the left but not right protected?

The paper's figures are beautifully made. Textbook quality.

Lines 192-193: Although not directly tied to the topic of fertility, it might be worth mentioning the role this is thought to play in yolk steroid deposition (Schwabl, H. (1993). Yolk is a source of maternal testosterone for developing birds. *Proceedings of the National Academy of Sciences USA* 90, 11446-11450; Groothuis, T.G.G., Müller, W., Von Engelhardt, N., Carere, C., & Eising, C. (2005). Maternal hormones as a tool to adjust offspring phenotype in avian species. *Neuroscience and Biobehavioral Reviews* 29, 329-352).

Line 220: But see: Arlt, D., Bensch, S., Hansson, B., Hasselquist, D., & Westerdahl, H. (2004). Observation of a ZZW female in a natural population: implications for avian sex determination. *Proceedings of the Royal Society of London B Biological Sciences* 271, S249-S251. doi:10.1098/rsbl.2003.0155

Lines 457-458: Increased relative to what? To a time more distant from ovulation?

Decision letter (RSOS-202274.R0)

Dear Miss Assersohn

The Editors assigned to your paper RSOS-202274 "Physiological factors influencing female fertility in birds" have now received comments from reviewers and would like you to revise the paper in accordance with the reviewer comments and any comments from the Editors. Please note this decision does not guarantee eventual acceptance.

Please submit your revised manuscript and required files (see below) no later than 21 days from today's (ie 19-Apr-2021) date. Note: the ScholarOne system will 'lock' if submission of the revision is attempted 21 or more days after the deadline. If you do not think you will be able to meet this deadline please contact the editorial office immediately.

on behalf of Professor Kevin Padian (Subject Editor)
openscience@royalsociety.org

Editor Comments to Author:

Thank you for your submission. As you will see the reviewers are generally favorable but have some rather different concerns. We ask you to address these carefully in your revision. Best wishes.

Reviewer comments to Author:

Reviewer: 1

Comments to the Author(s)

I would not consider this to be a comprehensive review. But based on the objectives listed at the beginning of the paper, I am not sure it is meant to be. There are several other papers on the topic in poultry in particular that are not included in this review.

In the sperm:egg interaction section of the review, I know of several papers that are not cited.

There is at least one theory of thought that I did not think was included and that is the interaction of the yolk with the IPVL in the non-germinal disc areas of the egg. I provided the authors some of those references but it has been a while since I have reviewed the literature on that topic and I would not consider this to be a comprehensive list.

The rest of my comments and suggestions are in the attached file.

Reviewer: 2

Comments to the Author(s)

I have not followed the avian ovarian physiology literature in several years, but I am knowledgeable in the area of reproductive ecology and behavior. So, I feel I should be able to provide some insight from that perspective. My overall impression of this manuscript is very favorable. It is an important topic, and the authors are correct that fertility problems have been poorly defined historically and that female fertility is a vastly understudied area. The authors seem to cover the topic thoroughly, so it is my opinion that this manuscript would make an

important and much-needed contribution to the field. With all that said, however, I am not convinced that Royal Society Open Science is the best venue for its publication, given the journal's own account of favoring papers that are of very broad, general interest. Indeed, I am not sure how broad and general the interest in physiological mechanisms of female bird fertility is. If the paper were on mechanisms of vertebrate or perhaps mammalian fertility, it might be of broader interest. But its current focus on female birds would seem to lend itself better to a bird journal, perhaps, or a journal more specifically oriented toward reproductive physiology, such as *Biology of Reproduction*.

As I indicated, however, I do feel that it is a worthwhile contribution, but some changes, as follows, would make it better. First, it is difficult to understand its overall organization. I wonder if a table of contents would be appropriate. I found it difficult to understand how each section tied in with the one before it and the one after it and then how they all fit together to address the central question: what causes infertility in female birds? One possibility would be to organize the paper around the anatomy of figure 2, walking the reader through each part of egg formation and how fertility can be affected in that part. Additionally, somewhere around line 41, it would be extremely helpful to list as many mechanisms for failed syngamy as possible in order to set up the rest of the paper. The paper could then be better organized around these. Currently, it seems vaguely organized around aspects of Figure 2, but a more deliberate breakdown of the specific ways syngamy can fail, perhaps in order of timing, would greatly facilitate the organization of the paper and therefore understanding it. Currently the paper reads like a long list of what we know about reproduction with little direct tie-in to failed syngamy, especially in any sort of chronological or other logical order.

I think about fertility from an adaptive perspective. Given the presence of a multitude of life-history trade-offs (particularly the trade-offs between number of offspring and quality of offspring within a single brood and also the trade-off between number of current offspring and number of future offspring), perhaps the term "failure" is inappropriate when describing when fertilization does not occur. There may be times when it benefits the female from a fitness perspective to avoid fertilization, even if it means the wasted time and energy that went into production of an infertile ovum or egg. Given that, at least in wild, altricial birds, far more investment occurs in feeding young than in forming the egg for the young, a female suddenly faced with an unexpected challenge (such as unexpected energy demands, predatory risks, or poor quality mate) might benefit by selectively blocking fertilization to avoid costly stages later during nestling rearing when conditions might not merit the investment. For example, the female may be constrained to laying 4 eggs, but challenging food conditions might mean that, this particular bout, 3 eggs might be more adaptive, so she can at least raise 3 good offspring rather than 4 poor ones. In this case, one way to save energy would be to adaptively prevent fertilization of one of the ova, giving rise to 4 eggs but only 3 mouths to feed. Ultimately, I think use of the terms "fail" and "succeed" when it comes to syngamy or fertilization is loaded and assumes fertilization is always adaptive, when there might be situations in which it is not adaptive.

Lines 408-410: Up to this point, the manuscript has mainly focused on what we know in domestic species, each time referring to how little is known in wild species. The reader is left wondering what IS known for wild species. So far, the manuscript mostly reads as a review in domestic species despite what seemed to be a focus early on on the importance of this topic to wild species.

Lines 686-687: It is this possibility specifically that I think of above all else when it comes to possible ways for fertilization to fail, and yet apparently nothing is known. Nonetheless, if the authors could go into greater detail, even if just speculation, I think it would resolve a lot of wondering by their readers.

In several places throughout the manuscript, the authors mention effects of aging on female fertility and mechanisms surrounding it. Although aging is relevant to an extent, it is unlikely to be particularly relevant in wild birds that never get very old. In fact, most wild birds may experience age-related increases in reproductive function during their short lives, and this may be due, in part, to priming by photostimulation (Sockman, K.W., Williams, T.D., Dawson, A., & Ball, G.F. (2004). Prior experience with photostimulation enhances photo-induced reproductive development in female European starlings: a possible basis for the age-related increase in avian reproductive performance. *Biology of Reproduction* 71, 979-986.)

Other comments:

Lines 28-29: Delete.

Line 40: Is this the most broadly used and accepted definition of infertility? If not, reword to say something like "We define infertility as . . ."

Line 87: It may be more technically difficult to study and understand female compared to male fertility.

Lines 133-136: Out of curiosity, how is the left but not right protected?

The paper's figures are beautifully made. Textbook quality.

Lines 192-193: Although not directly tied to the topic of fertility, it might be worth mentioning the role this is thought to play in yolk steroid deposition (Schwabl, H. (1993). Yolk is a source of maternal testosterone for developing birds. *Proceedings of the National Academy of Sciences USA* 90, 11446-11450; Groothuis, T.G.G., Müller, W., Von Engelhardt, N., Carere, C., & Eising, C. (2005). Maternal hormones as a tool to adjust offspring phenotype in avian species. *Neuroscience and Biobehavioral Reviews* 29, 329-352).

Line 220: But see: Arlt, D., Bensch, S., Hansson, B., Hasselquist, D., & Westerdahl, H. (2004). Observation of a ZZW female in a natural population: implications for avian sex determination. *Proceedings of the Royal Society of London B Biological Sciences* 271, S249-S251. doi:10.1098/rsbl.2003.0155

Lines 457-458: Increased relative to what? To a time more distant from ovulation?

===PREPARING YOUR MANUSCRIPT===

Please ensure that you include an acknowledgements' section before your reference list/bibliography. This should acknowledge anyone who assisted with your work, but does not

qualify as an author per the guidelines at <https://royalsociety.org/journals/ethics-policies/openness/>.

===PREPARING YOUR REVISION IN SCHOLARONE===

Author's Response to Decision Letter for (RSOS-202274.R0)

See Appendix B.

RSOS-202274.R1 (Revision)

Review form: Reviewer 1

Is the manuscript scientifically sound in its present form?

Yes

Are the interpretations and conclusions justified by the results?

Yes

Is the language acceptable?

Yes

Do you have any ethical concerns with this paper?

No

Have you any concerns about statistical analyses in this paper?

No

Recommendation?

Accept with minor revision (please list in comments)

Comments to the Author(s)

Personally I think your revisions have improved the paper greatly. I only have one minor suggestion before publishing. In every other case in the paper were the term "for example" was used, you provided an example. In one case you did not. See the Line 445 comment below.

Line 445 – This is a personal preference, but I do think it helps the readers if when the text says “although there are a few example...”, especially those new to the an area, make the information more useful to them, that some specific examples, maybe in parenthesis are provided.

Decision letter (RSOS-202274.R1)

Dear Miss Assersohn,

On behalf of the Editors, we are pleased to inform you that your Manuscript RSOS-202274.R1 "Physiological factors influencing female fertility in birds" has been accepted for publication in Royal Society Open Science subject to minor revision in accordance with the referees' reports. Please find the referees' comments along with any feedback from the Editors below my signature.

Please submit your revised manuscript and required files (see below) no later than 7 days from today's (ie 05-Jul-2021) date. Note: the ScholarOne system will 'lock' if submission of the revision is attempted 7 or more days after the deadline. If you do not think you will be able to meet this deadline please contact the editorial office immediately.

on behalf of Professor Kevin Padian (Subject Editor)
openscience@royalsociety.org

Associate Editor Comments to Author:

Thank you for your patience while we sought re-review. Unfortunately, only one of the original reviewers was available to assess your changes, and though a new reviewer had agreed to report, we regret that they were not able to do so in the end. With this in mind, we've opted to make a decision based on the feedback of the reviewer who did agree and report. Thank you for this contribution and we'll look forward to receiving the final version shortly.

Reviewer comments to Author:

Reviewer: 1

Comments to the Author(s)

Personally I think your revisions have improved the paper greatly. I only have one minor suggestion before publishing. In every other case in the paper where the term "for example" was used, you provided an example. In one case you did not. See the Line 445 comment below.

Line 445 - This is a personal preference, but I do think it helps the readers if when the text says "although there are a few examples...", especially those new to the area, make the information more useful to them, that some specific examples, maybe in parenthesis are provided.

===PREPARING YOUR MANUSCRIPT===

===PREPARING YOUR REVISION IN SCHOLARONE===

Author's Response to Decision Letter for (RSOS-202274.R1)

See Appendix C.

Decision letter (RSOS-202274.R2)

Dear Dr Assersohn,

I am pleased to inform you that your manuscript entitled "Physiological factors influencing female fertility in birds" is now accepted for publication in Royal Society Open Science.

on behalf of Prof Kevin Padian (Subject Editor)
openscience@royalsociety.org

Appendix A**ROYAL SOCIETY
OPEN SCIENCE****Physiological factors influencing female fertility in birds**

Journal:	Royal Society Open Science
Manuscript ID	RSOS-202274
Article Type:	Review
Date Submitted by the Author:	15-Dec-2020
Complete List of Authors:	Assersohn, Katherine; The University of Sheffield, Animal and Plant Sciences Brekke, Patricia; Institute of Zoology of the Zoological Society of London Hemmings, Nicola; The University of Sheffield, Animal & Plant Sciences
Subject:	physiology < BIOLOGY, evolution < BIOLOGY, health and disease and epidemiology < BIOLOGY
Keywords:	Hatching failure, Female fertility, Reproduction, Egg production, Sperm storage, Fertilisation
Subject Category:	Organismal and Evolutionary Biology

Author-supplied statements

Relevant information will appear here if provided.

Ethics

Does your article include research that required ethical approval or permits?:

This article does not present research with ethical considerations

Statement (if applicable):

CUST_IF_YES_ETHICS :No data available.

Data

It is a condition of publication that data, code and materials supporting your paper are made publicly available. Does your paper present new data?:

Yes

Statement (if applicable):

The dataset supporting this article has been uploaded as part of the supplementary material.

Conflict of interest

I/We declare we have no competing interests

Statement (if applicable):

CUST_STATE_CONFLICT :No data available.

Authors' contributions

This paper has multiple authors and our individual contributions were as below

Statement (if applicable):

K.A carried out the data collection, data analysis, created the figures and drafted the manuscript; P.B critically revised the manuscript; N.H participated in data collection, assisted in data analysis and critically revised the manuscript. All authors gave final approval for publication and agree to be held accountable for the work performed therein.

1 **Physiological factors influencing female fertility in** 2 **birds**

**Katherine Assersohn^{1*}, Patricia Brekke², Nicola Hemmings¹**

*¹Department of Animal and Plant Sciences, University of Sheffield, Sheffield, S10 2TN*

*²Institute of Zoology, Zoological Society of London, Regents Park, London, NW1 4RY*

**Abstract**

Fertility is fundamental to reproductive success, but not all copulation attempts result in a
fertilised embryo. Fertilisation failure is especially costly for females, and while there is a
growing appreciation for the considerable influence that female processes can have over
fertilisation, we lack a clear understanding of the causes of variation in female fertility across
taxa. Birds make a useful model system for fertility research, partly because their large eggs
are easily studied outside of the female's body, but also because of the wealth of data available
on the reproductive productivity of commercial birds. Here, we review the factors that
contribute to female infertility in birds, providing evidence that female fertility traits are being
understudied relative to male fertility traits, and that there is a bias in research effort towards
the study of Galliformes and captive (relative to wild) populations. We then highlight and
discuss the key physiological stages of the female reproductive cycle where fertility may be
compromised, and make recommendations for future research. We particularly emphasise the
need for studies to clearly differentiate between infertility and embryo mortality as causes of
hatching failure, and for information about non-breeding individuals to be more routinely
collected where possible. This review lays the groundwork for developing a clearer

understanding of the causes of female infertility, with important consequences for multiple
fields including reproductive science, conservation and commercial breeding.

*Key words:* hatching failure, female fertility, reproduction, egg production, sperm storage,
fertilisation

16 27 **I. Introduction** 17 18

[revised manuscript text omitted]

**Figure 3:** Schematic representation of a mature avian follicle. **I)** An avian follicle prior to
 ovulation; **II)** The ovum and follicle during ovulation, whereby a mature follicle ruptures at
 the stigma region, releasing the ovum. Sperm present in the infundibulum will begin to move
 towards the ovum in preparation for fertilisation, where they will penetrate the inner
 perivitelline layer (green); **III)** The ovum after fertilisation, the outer perivitelline layer
 (grey) has been laid down (which blocks further sperm entry). The inner perivitelline layer
 (green) has an abundance of sperm penetration holes around the germinal disc region where
 sperm have penetrated during fertilisation (see Fig. 2B).

*b) Ovulation*

Ovulation is a complex process under fine hormonal control [22]. It occurs when the largest
 mature yellow follicle (labelled F1 in Fig 2.) ruptures at the stigma region (Fig 3.) [25],
 releasing the ovum which is then captured by the infundibulum – the site of fertilisation. Unlike

mammals, the granulosa layer provides the main source of gonadal steroids [29], and ovulation
is initiated by the production of testosterone in the granulosa cells, which stimulates the release
of granulosa cell progesterone. Progesterone then creates a positive feedback response in the
hypothalamus which stimulates an increase in the secretion of gonadotropin-releasing
hormone, and consequently causes a surge of pituitary luteinising hormone [28,31,32]. Clock
genes expressed within granulosa cells after follicle selection are also thought to provide a
degree of circadian control over the timing of ovulation [28,33]. Proper regression of the post-
ovulatory follicle is thought to be required for managing the timing of ovulation and egg-laying
[22], and typically one ovum is released per day.

In broiler breeder hens, which have been selected for rapid growth at the expense of fertility,
double-yolk eggs are fairly common, and occur more frequently during the onset of
egg production [22,34]. Double-yolk eggs are associated with a greater incidence of embryo
mortality (at all stages of development) and are also more likely to be infertile [34], possibly
because ova are ovulated early and in an immature state. Ovulation order of double-yolk eggs
also affects the likelihood of fertilisation: in duck (*Anas platyrhynchos domesticus*) eggs, the
first yolk captured by the infundibulum has a higher probability of being fertilised [35]. This
may explain why double-yolk eggs commonly contain only one fertilised ovum [34]. Age,
nutrition (e.g. feed restriction) and changes in photostimulation are all thought to play a role in
the production of double-yolk eggs, the occurrence of which can also be increased via selection
[35], indicating a genetic component. In addition to double-yolks, the presence of multiple
germinal discs on a single yolk has been reported [36]. However, the cause and incidence of
such (and other) ovum abnormalities are unknown, as well as the implications they might have

[revised manuscript text omitted]

1286

Appendix B

We thank the editor and reviewers for their time and efforts spent reviewing our manuscript, and we thoroughly appreciate the comments and suggestions that have been provided. The reviewers offered us much food for thought, and we have extensively revised the manuscript based on their suggestions. We feel that the review is much improved as a consequence of this process. The changes we have made are detailed on a point-by-point basis, with each reviewer comment followed by our response in bold type. We also provide a version of the manuscript with all the changes highlighted to facilitate further review, and a clean version with all changes made but not highlighted. We hope you now consider our manuscript suitable for publication in Open Science.

Response to reviewer 1

Reviewer comment:

I would not consider this to be a comprehensive review. But based on the objectives listed at the beginning of the paper, I am not sure it is meant to be. There are several other papers on the topic in poultry in particular that are not included in this review.

Author reply:

We would like to thank Reviewer 1 for taking the time to read and comment on our paper – many of the suggestions made have been very useful, and have facilitated changes which we feel have significantly improved the manuscript. With respect to the reviewer’s point about the comprehensiveness of the review: our intention for this review was to integrate the most valuable and key insights from a broad range of fields. In this way we hope this review taps into a broader readership, encouraging greater cross-utility between different fields such as behavioural ecology, evolutionary biology, conservation science etc. We have therefore not been able to cite every relevant fertility paper across all these fields, however we do believe the review to be extensive and thorough, covering the most important aspects of female physiological function and fertility. To ensure our intentions for the review are not misunderstood, we have added a clarification to the introduction which describes its aims and scopes (lines 66-69). We have also included a number of extra citations in the text, including those that you have suggested. We would also like to mention that the quantitative analysis section was comprehensive, including every relevant fertility paper from a systematic Web of Science search, and these papers can all be found in the supplementary data provided.

Reviewer comment:

In the sperm:egg interaction section of the review, I know of several papers that are not cited. There is at least one theory of thought that I did not think was included and that is the interaction of the yolk with the IPVL in the non-germinal disc areas of the egg. I provided the authors some of those references but it has been a while since I have reviewed the literature on that topic and I would not consider this to be a comprehensive list.

Author reply:

Thank you for pointing this out, we have now included this in lines 633-639.

Reviewer comment:

Line 35 – 37: This is not an objective or definitive statement. Surely a review of the literature such as the current manuscript would be able to more better define the differences in male and female work. While one paper may state this, it does not necessarily make it true.

Author reply:

We have added additional lines here (lines 32-35) to further support the hypothesis that females have received less attention in avian fertility research. The intention for this line is to introduce the hypothesis that female fertility has been understudied, setting up the reader for the quantitative analysis which later provides evidence to support this.

Reviewer comment:

Line 54 – 56: Reproductive traits are only focused on in either the breeds used for table egg production or in some of the female broiler and turkey lines. Otherwise, the selection focus is on economically important traits (weight, weight gain, feed conversion, etc.) which are often inversely correlated with reproductive efficiency.

Author reply:

While it is certainly true that not all lines are selected for egg production traits, the focus of this point was to draw attention to the pervasive issues associated with egg production even in the layer lines where egg production has been specifically selected for. We later make the point that in some lines, economically important traits have been selected for at the expense of fertility (see lines 430 and 684). However, we have also altered this sentence (now line 53 and 55) to include the caveat that selection for efficient egg production has only been focussed on in certain lines.

Reviewer comment:

Line 83 – 84: The wording here makes it sound like scientists are ignoring the female fertility research but in reality it is more likely that male and male management factors are more significant factors that with correction influences the number of fertilized eggs more. Suggest rewriting this this sentence

Author reply:

We have included here some additional evidence that the pattern of a greater focus on male fertility is consistent across both wild populations, and captive populations – even when excluding the domestic chicken (which makes up over half of all the captive studies). This suggests that even beyond the studies made in poultry, male fertility is still focussed on more heavily. We have, however, altered this sentence here to avoid using the term 'bias' as per your suggestion, rewording it to the following: “there is a deficit of papers focusing on females (compared with males) within avian fertility research” on lines 86-87. We have also included a further statement about the practical advantages of studying male fertility traits for commercial species (lines 106 – 110), and reworded other lines of the review in keeping with the change in terminology.

Reviewer comment:

Line 92 – 93: Again, is bias the correct word to use here? More efforts on captive populations are probably overwhelmed by those studies that are carried out with poultry.

Author reply:

As above, we have altered the wording used here. We ensure that the new wording we have applied is consistent throughout the review when referring to this point.

Reviewer comment:

Line 100 – 108: While these points may be factors in the number of male to female fertility studies, it may just be the case, especially in poultry, that males mate with up to 10-15 females. So that management of the male with regards to fertility would affect the number of fertilised eggs more while having a larger economic impact.

Author reply:

It is definitely true that there are several practical advantages to studying sperm relative to ova, and we have further highlighted this within the text. However, the result we have found stands even when we remove domestic fowl (which make up half of all captive studies and is likely to be the main driver of any effects linked to commercial/economic impact) from our dataset. We have now included some additional evidence showing that even when removing domestic fowl, there are over twice as many papers focussed on males than on females. We also see this pattern mirrored in wild populations. We think this was an important point to raise. It suggests that even beyond any economic advantages to studying males for poultry science, this pattern is consistent across the entire field (see lines 100-104). That said, we do agree though that the historic economic and practical benefits for studying males in poultry has probably driven greater research, and this has likely contributed to the greater number of papers on males outside of poultry research as well, since we expect such advances to generate positive feedback in research effort. We had already made this point within the text, but have highlighted it further on lines 110-112.

Reviewer comment:

Line 140: Does the follicle or egg rupture?

Author reply:

It is the follicle that ruptures at the stigma region, releasing the ovum. To ensure this is clear we have adjusted the wording of this line (now line 160 – 162).

Reviewer comment:

Line 432 - 434: Bird size due to obesity as an impediment to successful mating?

Author reply:

Yes, thank you for pointing this out. This is now included (now line 450).

Reviewer comment:

Line 621: Another theory on the mechanism of preferential binding of sperm in the germinal disc region involved the discontinuous nature of the oolema which allow yolk to contact those areas of the IPVL. As a result of this, structural blocks from the yolk/IPVL interaction may prevent sperm from attaching. Bakst and Howarth 1977a; Robertson 1999. Also see Fertilization in Birds, 2000 Wishart and Horrocks in Fertilization in Protozoa and Metazoan Animals

Author reply:

Thank you, we have included a short statement regarding this hypothesis on line 633-639, including your suggested references in addition to some others.

Reviewer comment:

Line 701: How would it benefit? Up until this section the review has been backed by research studies. In these few sentences more opinion than fact are offered.

Author reply:

The previous sentence draws attention to our finding that far fewer studies have investigated fertility in non-poultry / wild species, which our earlier quantitative analysis finds evidence for. The intention here is to highlight that further studies in wild populations will rebalance our knowledge. But to highlight further the benefits to studying wild populations we have added an additional statement to expand on this (line 789 - 792)

Response to reviewer 2**Reviewer comment:**

I have not followed the avian ovarian physiology literature in several years, but I am knowledgeable in the area of reproductive ecology and behavior. So, I feel I should be able to provide some insight from that perspective. My overall impression of this manuscript is very favorable. It is an important topic, and the authors are correct that fertility problems have been poorly defined historically and that female fertility is a vastly understudied area. The authors seem to cover the topic thoroughly, so it is my opinion that this manuscript would make an important and much-needed contribution to the field. With all that said, however, I am not convinced that Royal Society Open Science is the best venue for its publication, given the journal's own account of favoring papers that are of very broad, general interest. Indeed, I am not sure how broad and general the interest in physiological mechanisms of female bird fertility is. If the paper were on mechanisms of vertebrate or perhaps mammalian fertility, it might be of broader interest. But its current focus on female birds would seem to lend itself better to a bird journal, perhaps, or a journal more specifically oriented toward reproductive physiology, such as *Biology of Reproduction*.

Author reply:

We would like to thank Reviewer 2 for the time taken to review our manuscript, and their positive comments and support of the need for this work. The suggestions made by Reviewer 2 have facilitated productive discussion, and the changes we have made as a consequence have significantly improved the paper. While we appreciate that historically, mammalian fertility is considered to be of more general interest, birds are one of the most intensely studied taxa in terms of reproduction and fertility. Birds are also far more diverse and more populous than mammals, and are incredibly popular study organisms within a variety of scientific fields. Furthermore, we believe that this review encourages greater integration between disciplines, bringing together an extensive range of literature that will interest a broad readership. Insights from birds – an important model species for fertility – will be of great value across taxa and we think it would be a shame to limit its readership to ornithologists alone. Royal Society Open Science specifically encourages reviews that generate new avenues for future work, which we have provided extensively throughout the review, as well as providing constructive critiques of fields, which we also focus on heavily. Consequently, we believe that Royal Society Open Science is the perfect home for this review.

Reviewer comment:

As I indicated, however, I do feel that it is a worthwhile contribution, but some changes, as follows, would make it better. First, it is difficult to understand its overall organization. I

wonder if a table of contents would be appropriate. I found it difficult to understand how each section tied in with the one before it and the one after it and then how they all fit together to address the central question: what causes infertility in female birds? One possibility would be to organize the paper around the anatomy of figure 2, walking the reader through each part of egg formation and how fertility can be affected in that part. Additionally, somewhere around line 41, it would be extremely helpful to list as many mechanisms for failed syngamy as possible in order to set up the rest of the paper. The paper could then be better organized around these. Currently, it seems vaguely organized around aspects of Figure 2, but a more deliberate breakdown of the specific ways syngamy can fail, perhaps in order of timing, would greatly facilitate the organization of the paper and therefore understanding it. Currently the paper reads like a long list of what we know about reproduction with little direct tie-in to failed syngamy, especially in any sort of chronological or other logical order.

Author reply:

This is an excellent suggestion. We have consequently restructured the review to pull focus back onto the specific mechanisms of female infertility in birds. As you suggested, we have organised the paper to follow roughly the anatomy of Figure 2. The introduction and systematic analysis of the literature sections have remained as they were, but section III is now restructured with the title of this section and the title of the subheadings pulling more focus onto the mechanisms of fertilisation failure. We feel this has much improved the entire manuscript, with the organisation having a more succinct and understandable flow. We also were able to provide a broad list of the mechanisms of infertility within the introduction (line 72 – 74), as you suggested, and this sets the reader up nicely for the rest of the review since this list now mirrors the overall structure of section III. We also found that by restructuring the review we have naturally incorporated many of your other suggestions and comments. We decided not to include a contents table given that this does not seem fitting with the general style of Royal Society Open Science, but have provided a table of the new structure for section III below for your information:

III. What causes fertilisation failure in female birds?

- 1 Failure during egg formation**
 - 1.1 Hormonal factors**
 - 1.2 Disease and immune factors**
 - 1.3 Environmental factors**
 - 1.3.1 Diet**
 - 1.3.2 Stress**
 - 1.3.3 Pollution**
- 2 Failure during ovulation**
 - 2.1 Hormonal factors**
 - 2.2 Disease and immune factors**
- 3 Failure to obtain sperm**
 - 3.1 Copulation**
 - 3.2 Timing of insemination**
 - 3.3 Vaginal sperm selection**
- 4 Failure to maintain and transport sperm**
 - 4.1 Sperm storage tubule function**
 - 4.2 Sperm release and transport**

5 Failure of sperm-egg fusion

5.1 Sperm-egg interactions

5.2 Syngamy

Reviewer comment:

I think about fertility from an adaptive perspective. Given the presence of a multitude of life-history trade-offs (particularly the trade-offs between number of offspring and quality of offspring within a single brood and also the trade-off between number of current offspring and number of future offspring), perhaps the term "failure" is inappropriate when describing when fertilization does not occur. There may be times when it benefits the female from a fitness perspective to avoid fertilization, even if it means the wasted time and energy that went into production of an infertile ovum or egg. Given that, at least in wild, altricial birds, far more investment occurs in feeding young than in forming the egg for the young, a female suddenly faced with an unexpected challenge (such as unexpected energy demands, predatory risks, or poor quality mate) might benefit by selectively blocking fertilization to avoid costly stages later during nestling rearing when conditions might not merit the investment. For example, the female may be constrained to laying 4 eggs, but challenging food conditions might mean that, this particular bout, 3 eggs might be more adaptive, so she can at least raise 3 good offspring rather than 4 poor ones. In this case, one way to save energy would be to adaptively prevent fertilization of one of the ova, giving rise to 4 eggs but only 3 mouths to feed. Ultimately, I think use of the terms "fail" and "succeed" when it comes to syngamy or fertilization is loaded and assumes fertilization is always adaptive, when there might be situations in which it is not adaptive.

Author reply:

This is an interesting point, however we believe that it goes somewhat beyond the scope of the current review to consider the adaptive potential of infertility. Here, we are solely concerned with physiological fertility problems encountered by female birds, making the term fertilisation 'failure' appropriate in this case. Furthermore, the potential for infertility to be adaptive is currently rather speculative, with (to our knowledge) no current research indicating that infertility itself could be advantageous, and no known mechanisms by which it could occur. We believe there is a strong argument that selective embryo mortality (i.e. rejection of the egg after laying) could be adaptive, and indeed this does occur in some species with some females choosing not to incubate certain eggs/clutches. However, we do not believe a block to fertilisation is likely to be either physiological feasible or adaptive. That is because, once ovulation has occurred, the female has no choice but to lay the resulting egg. A mechanical mechanism to block fertilisation of an ovum would therefore not ameliorate any costs of reproduction and so be unlikely to evolve. Theoretically, if a female could selectively fail to produce an egg in the first place, this could be adaptive since it avoids the initial heavy cost of egg production and laying. However, in many species, females lay eggs even when they do not have access to sperm, suggesting that control over egg production itself is limited. A final theoretical mechanism would be to avoid copulation itself, although this would not be possible for species where forced copulations are common. Females could alternatively physically reject sperm before it reaches the site of fertilisation, though this would not be adaptive for species that lay eggs even without the presence of sperm, and would not ameliorate the costs associated with copulation either. Given that, we do not believe

there is currently a strong argument for infertility to be adaptive, and so for the purposes of this review, we believe that the term fertilisation 'failure' is acceptable.

Reviewer comment:

Lines 408-410: Up to this point, the manuscript has mainly focused on what we know in domestic species, each time referring to how little is known in wild species. The reader is left wondering what is known for wild species. So far, the manuscript mostly reads as a review in domestic species despite what seemed to be a focus early on on the importance of this topic to wild species.

Author reply:

We agree that it is surprising and somewhat frustrating that such little knowledge exists for wild populations. Indeed, it is partly our intention to highlight this within the review. We have included the most relevant and key insights gained from wild species, which admittedly is dwarfed by what is known in domestic birds. We agree however, that the inclusion of this information was not well balanced throughout the review under the previous structure. As per your recommendation, we have restructured the review, and one of our goals here was also to better highlight the findings for wild species within this new structure. Since the majority of research into fertility in wild species has focussed on the influence of environmental factors (such as heat stress/ climate change and pollution), we have drawn better attention to this by creating a separate 'environmental factors' subheading early in the review. We would also like to draw your attention to the analysis where we have further highlighted that there is a deficit of fertility papers on wild populations, and we also return to this point within the conclusions and future directions section.

Reviewer comment:

Lines 686-687: It is this possibility specifically that I think of above all else when it comes to possible ways for fertilization to fail, and yet apparently nothing is known. Nonetheless, if the authors could go into greater detail, even if just speculation, I think it would resolve a lot of wondering by their readers.

Author reply:

We agree that it is somewhat surprising that so little is known about how ovum quality influences fertility in birds. To address your suggestion, we have included some more speculation as to the influence of ovum quality on fertility, specifically drawing more parallels with what is known in mammals, and suggesting similar mechanisms could occur in birds. We believe this section is now much improved due to this inclusion.

Reviewer comment:

In several places throughout the manuscript, the authors mention effects of aging on female fertility and mechanisms surrounding it. Although aging is relevant to an extent, it is unlikely to be particularly relevant in wild birds that never get very old. In fact, most wild birds may experience age-related increases in reproductive function during their short lives, and this may be due, in part, to priming by photostimulation (Sockman, K.W., Williams, T.D., Dawson, A., & Ball, G.F. (2004). Prior experience with photostimulation enhances photo-induced reproductive development in female European starlings: a possible basis for the age-related increase in avian reproductive performance. *Biology of Reproduction* 71, 979-

986.)

Author reply:

Thank you, yes this is an interesting point. While we do see an initial increase in reproductive success in some seasonally reproducing birds, the general and gradual decline in fertility with age is common across species. It is usually observed in birds as an increase in the length of inter-clutch intervals, smaller clutch sizes and a general increase in the number of failed reproductive attempts (Ellison & Ottinger, 2014). You are right in that this is primarily of concern to captive populations of both commercial and non-commercial birds which do tend to live longer than many wild species and also be pushed to be reproductively active for longer too. However, ageing in wild species has been shown to also be widespread across birds and mammals (e.g. Nussey et al., 2013; Robery et al., 2015), and it is also a very important consideration for some endangered species, (such as the whooping crane (Brown et al., 2019)) and for managed wild populations where management interventions (such as supplemental feeding) influences patterns of ageing. We have consequently included a short discussion here (line 340 -353), emphasising that fertility can initially increase with photo-experience. We have included your references here as well as the notable example of fertility senescence in whooping cranes, and the contradictory example of long-lived seabirds who experience apparently no reproductive decline throughout their lives.

Reviewer comment:

Lines 28-29: Delete.

Author reply:

Thank you, we have deleted this line.

Reviewer comment:

Line 40: Is this the most broadly used and accepted definition of infertility? If not, reword to say something like "We define infertility as . . . "

Author reply:

Thank you, we have corrected this.

Reviewer comment:

Line 87: It may be more technically difficult to study and understand female compared to male fertility.

Author reply:

We have further highlighted the technical and practical advantages to studying male fertility vs female fertility (lines 106-110).

Reviewer comment:

Lines 133-136: Out of curiosity, how is the left but not right protected?

Author reply:

This is a really interesting question which unfortunately doesn't appear to have a clear answer. Briefly - both the left and the right ovary begin developing together but then asymmetric development is observed after around 6 days of incubation. Development of the left ovary is promoted by asymmetrical gene expression and migration of primordial germ cells into the left gonad during embryogenesis. Expression of pituitary homeobox 2 (*PITX2*) within the left ovary suppresses oestrogen receptors in the right ovary, resulting in

elevated concentrations of oestrogen in the left ovary relative to the right. The anti-Müllerian hormone (AMH) (the expression of which is regulated by the transcription factor SF1) is then thought to initiate apoptosis of the right Müllerian duct, causing it to regress. As briefly mentioned in the manuscript, it is widely thought that the elevated concentrations of oestrogen in the left ovary suppress the anti-Müllerian hormone receptor, and this prevents the left ovary from regressing. Unfortunately, the molecular mechanisms underpinning this have not been fully defined, and while many papers do find evidence for apoptosis in the right ovary (e.g. Ukeshima 1996, Grzegozewska 2012, Shaikat et al., 2018 etc), there is at least one case where apoptosis was not evident (De Melo Bernardo et al., 2015). We believe including full details of the molecular mechanisms governing this process goes beyond the scope of this paper, however we have added a clarification here that this is still an area of active research.

Reviewer comment:

The paper's figures are beautifully made. Textbook quality.

Author reply:

Thank you for this comment!

Reviewer comment:

Lines 192-193: Although not directly tied to the topic of fertility, it might be worth mentioning the role this is thought to play in yolk steroid deposition (Schwabl, H. (1993). Yolk is a source of maternal testosterone for developing birds. *Proceedings of the National Academy of Sciences USA* 90, 11446-11450; Groothuis, T.G.G., Müller, W., Von Engelhardt, N., Carere, C., & Eising, C. (2005). Maternal hormones as a tool to adjust offspring phenotype in avian species. *Neuroscience and Biobehavioral Reviews* 29, 329-352).

Author reply:

Thank you for bringing this up, and it is certainly an interesting phenomenon, however here we primarily discuss the influence of gonadal steroids only as it pertains to maternal fertility (in this case ovulation), not of how maternal hormones later go on to influence embryo development. We have tried to steer away from a discussion of embryo development as much as possible given how common it is for researchers to confuse infertility and embryo mortality with each other. We believe a discussion of this might muddy the waters in terms of fertility, so have chosen to leave this out.

Reviewer comment:

Line 220: But see: Arlt, D., Bensch, S., Hansson, B., Hasselquist, D., & Westerdahl, H. (2004). Observation of a ZZW female in a natural population: implications for avian sex determination. *Proceedings of the Royal Society of London B Biological Sciences* 271, S249-S251. doi:10.1098/rsbl.2003.0155

Author reply:

Thank you, we have now included this reference (now line 718)

Reviewer comment:

Lines 457-458: Increased relative to what? To a time more distant from ovulation?

Author reply:

We have reworded and improved the clarity of this section which is now on lines 479 – 484.

Appendix C

We are thrilled at the editor's decision to accept our manuscript for publication. We would like to thank the editors for their time and efforts spent attempting to obtain a second reviewer to assess the changes we have made. We appreciate the decision to move forward based on the feedback from reviewer 1, and further thank reviewer 1 for the time taken to review our manuscript for a second time. We have addressed the additional suggestion made by reviewer 1 below and have attached both a clean and marked up version of the manuscript with the changes highlighted. Thank you again for accepting our paper for publication in Open Science.

Kind regards,

Katherine Assersohn, Patricia Brekke and Nicola Hemmings

Response to reviewer 1

Reviewer comment:

Personally I think your revisions have improved the paper greatly. I only have one minor suggestion before publishing. In every other case in the paper where the term "for example" was used, you provided an example. In one case you did not. See the Line 445 comment below.

Line 445 – This is a personal preference, but I do think it helps the readers if when the text says "although there are a few examples...", especially those new to the area, make the information more useful to them, that some specific examples, maybe in parentheses are provided.

Author reply:

We agree the paper is much improved and this is largely thanks to your helpful feedback, which we are extremely grateful for. Thank you for the additional comment, we agree and have addressed this in the manuscript by adding several examples in parentheses as you suggest, and we have included two additional references.